# Experimental Design for Multi-Channel Imaging via Task-Driven Feature Selection

**Stefano B. Blumberg**[1,2], **Paddy J. Slator**[2,3], **Daniel C. Alexander**[2]

[1]Centre for Artificial Intelligence, Department of Computer Science, University College London
[2]Centre for Medical Image Computing, Department of Computer Science, University College London
[3]Cardiff University Brain Research Imaging Centre and School of Computer Science, Cardiff University
`stefano.blumberg.17@ucl.ac.uk`

## Abstract

This paper presents a data-driven, task-specific paradigm for experimental design, to shorten acquisition time, reduce costs, and accelerate the deployment of imaging devices. Current approaches in experimental design focus on model-parameter estimation and require specification of a particular model, whereas in imaging, other tasks may drive the design. Furthermore, such approaches often lead to intractable optimization problems in real-world imaging applications. Here we present a new paradigm for experimental design that simultaneously optimizes the design (set of image channels) and trains a machine-learning model to execute a user-specified image-analysis task. The approach obtains data densely-sampled over the measurement space (many image channels) for a small number of acquisitions, then identifies a subset of channels of prespecified size that best supports the task. We propose a method: TADRED for TAsk-DRiven Experimental Design in imaging, to identify the most informative channel-subset whilst simultaneously training a network to execute the task given the subset. Experiments demonstrate the potential of TADRED in diverse imaging applications: several clinically-relevant tasks in magnetic resonance imaging; and remote sensing and physiological applications of hyperspectral imaging. Results show substantial improvement over classical experimental design, two recent application-specific methods within the new paradigm, and state-of-the-art approaches in supervised feature selection. We anticipate further applications of our approach. Code is available: Code Link.

## 1 Introduction

Experimental design seeks a sampling scheme or design $D = \{\mathbf{d}^1, ..., \mathbf{d}^C\}$, where each $\mathbf{d}^i$, $i = 1, ..., C$, is a combination of experimental variables that are under the control of the experimenter, that provides data optimally informative for some criteria or task Antony (2003); Pukelsheim (2006). The experimental outcome (measured data) of design $D$ is a matrix $X_D \in \mathbb{R}^{n \times C}$ with $C$ corresponding measurements from each of $n$ samples. The optimal choice of design depends on the experimental task, which we express as a function $\mathcal{T}$ that maps $X_D$ to a corresponding matrix $Y$ of labels. Experimental design optimization seeks the design that maximizes the ability to perform the task, subject to constraints of time or cost, i.e.

$$D^* = \arg\min_D L(\mathcal{T}(X_D), Y), \quad \text{subject to} \ |D| = C \tag{1}$$

where $L$ is a loss function. Here we limit cost simply to the size $C$ of $D$; $\mathcal{T}$ can be any task, but often in imaging involves estimating/mapping model parameters e.g. via gradient-descent model-fitting in every pixel/voxel, as in Alexander (2008); Cercignani & Alexander (2006), or machine learning as in Gyori et al. (2022); Waterhouse & Stoyanov (2022).

In imaging, as illustrated in figure 1a, $X_D$ is typically a collection of $n$ pixels or voxels with $C$ channels (e.g. RGB images have $C = 3$). The choice of $\mathbf{d}^i \in D$ controls the contrast in channel $i$ and is global to the whole channel. Compact (small $C$) but informative designs are often critical in reducing acquisition or development costs in real-world applications. Examples include acquiring magnetic resonance imaging (MRI) contrasts, e.g. to estimate and map microstructural tissue

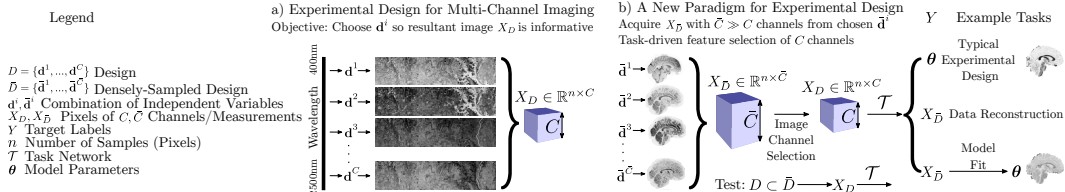

Figure 1: a) An example of experimental design for imaging. In remote sensing hyperspectral imaging (see table 3), each observed wavelength $\mathbf{d}^i$ is chosen by the experimenter. The outcome of each $\mathbf{d}^i$ is a grayscale image - a channel of the resultant data $X_D$ (RGB has 3 channels). b) The new paradigm for experimental design illustrated for qMRI. First, obtain image data $X_{\bar{D}}$ with a large number of $\bar{C}$ channels. Next, train a user-chosen task network, which drives design optimization to select $C < \bar{C}$ channels – we propose TADRED for this. We consider three distinct example tasks in experiments.

properties within the time a patient can stay still in a scanner Alexander (2008), or manufacturing affordable hyperspectral imaging devices including a few well-chosen spectral filters, e.g. for estimating tissue oxygenation Waterhouse & Stoyanov (2022).

Standard approaches for experimental design typically optimize $D$ over a continuous space, for the task of model parameter estimation. For example, a classical approach still widely deployed uses the Fisher matrix Montgomery (2001), whilst more recent approaches use the paradigm of sequential Bayesian experimental design Blau et al. (2022); Foster et al. (2021); Ivanova et al. (2021). Both require a priori model choice, limiting consideration to model-based tasks, and even specific model-parameter choices or assumptions on their prior distribution. Moreover, such approaches rapidly become computationally intractable as the dimension of the optimization increases.

Here we suggest a new task-driven paradigm for experimental design for real-world imaging applications, illustrated in figure 1b, that does not require a priori model specification and replaces high-dimensional continuous search with a subsampling problem. First, the paradigm requires training data $X_{\bar{D}}$ with $\bar{C}$ channels/measurements acquired using a design $\bar{D}$ that densely samples the measurement space. Secondly the paradigm selects a subset of size $C \ll \bar{C}$ image channels from $X_{\bar{D}}$ (optimizing the design and choosing $X_D \subset X_{\bar{D}}$), coupled with the training of a high-performing neural network that executes the task $\mathcal{T}$ driving the experimental design. Thus, the new paradigm replaces the optimization in equation 1 with:

$$D^*, \mathcal{T}^* = \underset{D,\mathcal{T}}{\arg\min} \, L(\mathcal{T}(X_D), Y) \text{ subject to } D \subset \bar{D}. \tag{2}$$

In this paradigm, the task must be specified a priori, but may go beyond standard model-based tasks that drive classical/Bayesian experimental design, to include 'model free' tasks such as missing data reconstruction. The training data requires only a small number of subjects/samples, so may use specialized hardware, lengthy acquisitions, or even simulations. In practice, such acquisitions are often made during early development phases of imaging technologies to explore the range of sensitivity, which informs the choice of, and often provides, $\bar{D}$. The paradigm we propose formalizes the exploitation of such data in experimental design for downstream systems designed for wide deployment and directly supports the use of deep learning for $\mathcal{T}$.

In the new paradigm, the experimental design problem becomes similar to supervised feature selection, where the $\bar{C}$ image channels of $X_{\bar{D}}$ are considered features. In supervised feature selection, state-of-the-art approaches Wojtas & Chen (2020); Lee et al. (2022) couple feature selection with task optimization, however the structure of the data in typical supervised feature selection problems differs from those in experimental design for imaging. Feature selection algorithms typically assume most features are uninformative and the task is to 'identify a small, highly discriminative subset' Kuncheva et al. (2020) e.g. genes associated with drug response from the entire genome. In experimental design for imaging, however, most channels individually offer similar amounts of information to support task performance, since they view the same scene/sample but with often-subtle differences in contrast (see e.g. figure 6). Design optimization seeks a compact combination that covers all important aspects.

Therefore we propose TADRED, a novel method for TAsk-DRiven Experimental Design in imaging. TADRED couples feature scoring and task execution in consecutive networks. The scoring and subsampling procedure enables efficient identification of subsets of complementarily informative

channels jointly with training a high-performing network for the task. TADRED also gradually reduces the full set of samples stepwise to obtain the subsamples, which improves optimization.

Key contributions are:

1. A new coupled subsampling-task paradigm (feature selection) for experimental design in imaging.
2. TADRED: a novel approach for supervised feature selection tuned specifically for experimental design in imaging. TADRED performs task-based image channel selection.
3. A demonstration of our approach on six datasets/tasks in both clinically-relevant MRI and remote sensing and physiological applications in hyperspectral imaging. TADRED outperforms (i) Classical experimental design, (ii) Recent application-specific published results, (iii) State-of-the-art approaches in supervised feature selection.

## 2 RELATED WORK

**Approaches in Experimental Design** A typical task in experimental design is to optimize the design $D$ for estimating model parameters. The most widely used classical approach in imaging uses the Fisher information matrix Pukelsheim (2006). However, for non-linear models, the optimization requires pre-specification of parameter values of interest, leading to circularity, e.g. the standard design for VERDICT model with primary application in prostate cancer detection and classification Panagiotaki et al. (2015a) (used as a baseline in table 1) is computed by optimizing the Fisher-matrix for one specific combination of parameter values, despite aiming to highlight contrast in those parameters throughout the entire prostate. Approaches in the sequential Bayesian experimental design paradigm Blau et al. (2022); Foster et al. (2021); Ivanova et al. (2021) reduce this circularity by optimizing over combinations or ranges of parameter values. Recently Blau et al. (2022) also implemented an experimental design optimization in a discrete space and obtained state-of-the-art performance and deployment time, by using reinforcement learning to map history of designs and outcomes to the next design. However, the tasks driving experimental design in imaging are often 'model free' supervised tasks such as missing data reconstruction (tables 2, 3) to recover missing image channels. Classical Fisher-matrix experimental design or sequential Bayesian techniques do not apply in such problems. Furthermore, the sequential Bayesian techniques have been deployed on only small-scale experiments with simulated data e.g. a simple localization problem for two sources. For example, experiments in Blau et al. (2022) have $C \leq 2$ and $D \in \mathbb{R}^{\dim}, \dim \leq 6$. In contrast, e.g. the real-world experiment in table 1 has $C \in \{110, 55, 28, 14\}$ and $D \in \mathbb{R}^{7 \cdot C}$. Preliminary experiments suggest the application of these approaches to the high dimensional problems is not computationally tractable with the published code/methods. These issues motivate the reformulation of the experimental design paradigm and the introduction of TADRED. Appendix-E is a broader review of experimental design for quantitative MRI (qMRI) and hyperspectral imaging.

**Supervised Feature Selection** operates either at the instance level e.g. identifying different salient parts of different images; or at the population level by selecting across all the instances. In imaging, each combination of acquisition parameters $\mathbf{d}^i \in D$ is global across all image pixels/voxels, so channel-selection for experimental design must be population-wide. Recursive feature elimination (RFE) / backward selection Guyon et al. (2002); Scikit-Learn (2023); Kohavi & John (1997) are frameworks that seek the most informative set of features among a superset to inform a model or task. They work by eliminating the least informative features stepwise to reach a prespecified feature-set size. 'Feature Importance Ranking for Deep Learning' (FIRDL) Wojtas & Chen (2020), 'Self-Supervision Enhanced Feature Selection with Correlated Gates' (SSEFS) Lee et al. (2022) are considered state-of-the-art in feature selection, outperforming both classical (e.g. RFE) and recent approaches outlined in appendix-E. Both techniques are specifically designed to 'identify a small, highly discriminative' subset Kuncheva et al. (2020) of features from a larger group of mostly uninformative features. SSEFS, in a first step, uses a probabilistic approach to search for this subset, whilst also exploiting the presence of correlated subsets for enhanced performance. A second step then trains a network on the chosen subset to execute the task. FIRDL instead has a complex optimization procedure involving exploration-exploitation stochastic local search. SSEFS and FIRDL are detailed in appendix A and are baselines in later experiments.

In contrast to typical feature selection problems, most candidate choices in experimental design are informative: few, if any, features are uninformative so no single small discriminative set exists. SSEFS's first step seeks groups of correlated features, which is less useful in experiment design, as

most image channels correlate strongly (examples in figure 5). FIRDL incorporates global information by performing multiple evaluations on different feature combinations. However, FIRDL's search for a discriminative subset is inappropriate in the experimental design application; its multiple evaluations of the task-execution network are redundant and result in covariate shift and overfitting.

Nevertheless, TADRED builds upon the basic principles of task-driven feature selection, which is the foundation of FIRDL and SSEFS's success. TADRED adopts the same dual-network architecture, but with a different optimization procedure tailored to the experimental design problem. Specifically, TADRED's implements a novel combination of the dual selection/task network optimization within the paradigms of RFE/backwards selection. As such, it adopts a comparatively simple scoring procedure, which avoids the complicated and suboptimal joint optimization FIRDL/SSEFS require to search for a distinctively discriminative subset. TADRED's end-to-end dual networks avoids FIRDL's multiple evaluations on different feature combinations, and TADRED's passing of information through the optimization procedure improves on both SSEFS and FIRDL.

Finally, PROSUB Blumberg et al. (2022) (baseline in table 2) is a previous attempt to equate experimental design with feature selection and also uses RFE. It uses a customized neural architecture search at every step and was designed specifically to address a measurement-selection problem in qMRI (data in table 2) where it achieves state-of-the-art performance. However, the technique does not naturally generalize to other tasks, which is a key motivation for TADRED. TADRED avoids PROSUB's cumbersome neural architecture search and implements instead a novel four-phase procedure in each step, which keeps the gradient updates smooth and allows feature selection at each step. Also, beyond standard RFE, TADRED efficiently passes information from the optimization on larger feature sets to smaller sets by passing information on the network weights across the steps, unlike PROSUB. These advances combine to enhance substantially the performance, portability and generalizability of the algorithm across diverse experimental design problems.

# 3 TADRED: TAsk-DRiven Experimental Design for Imaging

TADRED presents a novel approach to supervised feature selection, tailored to the particularities of the experimental design problem in imaging, and aims to solve equation 2. Section 3.1 describes an outer loop of the procedure, which is inspired by classical paradigms Kohavi & John (1997); Guyon et al. (2002), that gradually eliminates elements from the densely-sampled design $\bar{D}$ in $t = 1, ..., T$ steps to obtain designs $\bar{D} = D_1 \supset ... \supset D_T$. This corresponds to performing supervised feature selection for decreasing sizes $\bar{C} = C_1 > ... > C_T$, where $\{C_t\}_{t=1}^{T}$ are chosen by the user a priori. Then section 3.2 outlines an inner loop for training with fixed $1 \leq t \leq T$. Inspired by recent supervised feature selection advances Imrie et al. (2022); Wojtas & Chen (2020), TADRED trains two coupled networks at each step: a scoring network $\mathcal{S}_t$, which scores individual elements of $X_{\bar{D}}$ for importance to inform the subsampling, and a task network $\mathcal{T}_t$, which performs the task driving the design, i.e. estimates $Y$ from chosen feature subset $X_{D_t} \subset X_{\bar{D}}$. The training procedure is split into four phases that allows feature selection at each step and is inspired by Karras et al. (2018); Blumberg et al. (2022) which produced enhanced optimization. The full procedure is outlined in algorithm 2.

## 3.1 Outer Loop

Across steps $t = 1, ..., T$ we consider decreasing feature set sizes $\bar{C} = C_1 > C_2 > ... > C_T$ and perform supervised feature selection at each step in an inner loop (see section 3.2). Reducing feature set sizes stepwise aids the optimization procedure compared to e.g. training on all features then subsampling all at once (see table 5). The procedure passes information from the optimization on larger feature sets to smaller sets. Finally, the stepwise procedure efficiently produces a set of optimized designs (as is typical in supervised feature selection see e.g. Wojtas & Chen (2020) and also in Waterhouse & Stoyanov (2022)), which can be useful for post-hoc selection of design size to balance economy (small C) with task performance. Whilst iterative subsampling also increases computational time, this is comparable to other supervised feature selection approaches (appendix D).

## 3.2 Inner Loop: Four-Phase Deep Learning Training

At step $1 \leq t \leq T$ of the outer loop the inner loop constructs (i) a binary mask $\mathbf{m}_t \in \{0, 1\}^{\bar{C}}, ||\mathbf{m}_t||_0 = C_t$ to subsample the features; (ii) a weight vector for the features $\bar{\mathbf{s}}_t \in \mathbb{R}_+^{\bar{C}}$;

(iii) a trained network $\mathcal{T}_t$ to perform the task, which corresponds to solving the optimization problem:

$$\underset{\mathbf{m}_t,\ \mathcal{T}_t,\ \bar{\mathbf{s}}_t}{\text{minimize}}\ L(\mathcal{T}_t(X_{D_t} \odot \bar{\mathbf{s}}_t), Y),\ \ \text{subject to}\ ||\mathbf{m}_t||_0 = C_t,$$

$$\text{where } X_{D_t} = \mathbf{m}_t \odot X_{\bar{D}} + (\mathbf{1}_{\bar{C}} - \mathbf{m}_t) \odot X_{\bar{D}}^{\text{fill}}, \tag{3}$$

the $\odot$ operation is element-wise dot product which follows broadcasting rules when inputs have mismatched dimensions, $||\cdot||_0$ is the $L^0$ norm, $\mathbf{1}_{\bar{C}}$ is a vector with $\bar{C}$ ones, and 'feature fill' $X_{\bar{D}}^{\text{fill}} \in \mathbb{R}^{\bar{C}}$ is a hyperparameter that fills the removed features (we take the data median, see appendix C.2). The weight vector $\bar{\mathbf{s}}_t$ contains feature scores, which the training procedure uses to remove low-scoring features by setting corresponding values of the mask $\mathbf{m}_t$ to 0.

**Scoring, Subsampling, and Task Execution** The core of the training procedure uses the forward/backward pass in algorithm 1. The full procedure in algorithm 2 uses the forward/backward pass to update feature scoring gradually in tandem with improving label prediction.

The procedure aims to learn a meaningful sample-independent feature score to rank the features. In practice, deep-learning training is performed in batches and not across the whole data. Therefore we first learn a sample-dependent feature score $\sigma(\mathcal{S}_t(X_{\bar{D}})) = \tilde{s} \in \mathbb{R}_+^{n \times \bar{C}}$, where $\mathcal{S}_t$ is a neural network and $\sigma : \mathbb{R} \to [0, \infty)$ is an activation function to ensure positive scores (we take $\sigma = 2 \cdot \text{sigmoid}$ and at initialization $\sigma(0) = 1$). We then compute a sample-independent score $\bar{\mathbf{s}}_t \in \mathbb{R}_+^{\bar{C}}$ as an average of $\tilde{s}$ across the $n$ samples in $X_D$. We also compute a combined score that aids task execution

$$\mathbf{s} = \alpha \odot \tilde{s} + (1 - \alpha) \odot \bar{\mathbf{s}}_t,\ \ \alpha \in [0, 1], \tag{4}$$

which balances the current learned sample-dependent score with a fixed global estimate of the sample-independent score and allows smooth integration between the two. The mix parameter $\alpha$ is set in the optimization procedure to shift the balance from sample-dependent to sample-independent scores.

We use a mask $\mathbf{m}_t \in [0, 1]^{\bar{C}}$ to subsample the features $X_{D_t} = \mathbf{m}_t \odot X_{\bar{D}} + (\mathbf{1}_{\bar{C}} - \mathbf{m}_t) \odot X_{\bar{D}}^{\text{fill}}$, and replace the removed features with default values $X_{\bar{D}}^{\text{fill}}$ to retain the shape of the data structures throughout training. Rather than learning the mask $\mathbf{m}_t$ end-to-end e.g. using a sparsity term/prior as in Lee et al. (2022), we modify elements of $\mathbf{m}_t$ during our training procedure. This is important to enable the outer loop of the procedure to output candidate designs at each step.

We now estimate the target $Y$ with $\widehat{Y} = \mathcal{T}_t(\mathbf{s} \odot X_{D_t})$ from the subsampled data weighted feature-wise by the score, then calculate the loss $L(\widehat{Y}, Y)$. This weighting allows gradients to flow end-to-end.

**Training Procedure** The key challenges in the design of the training procedure in the inner loop are how to (i) obtain meaningful global sample-independent scores $\bar{\mathbf{s}}_t$ from learnt sample-dependent scores $\tilde{s}_t$, (ii) differentiate through a masking operation to compute $\mathbf{m}_t$. TADRED's four-phase procedure, inspired by Karras et al. (2018); Blumberg et al. (2022) gradually modifies the neural network structure during deep learning training, moving from learning a simpler task (learning sample-dependent scores and retaining most features) to a more complex task (learning sample-independent scores and removing more features) by linear interpolation of network components. This improves optimization over directly learning the more difficult task. Thus we address (i) by first learning $\tilde{s}$ and then progressively reducing the final score to the average of $\tilde{s}$ its average i.e. $\mathbf{s} = \bar{\mathbf{s}}_t$ (in algorithm 2 phase 2), and (ii) by progressively setting elements of $\mathbf{m}_t$ to zero i.e., during training, mask elements are real valued but gradually reduce to binary values (in phase 3).

The training procedure is different for the first outer loop step $t = 1$ compared to steps $t \geq 2$. This is because for step $t = 1$ we train on all $\bar{C}$ features and do not have information from previous steps and for steps $t \geq 2$ we perform supervised feature selection for user-chosen $C_t$ (solve equation 3) and training is initialized from step $t - 1$. We describe each step with reference to algorithm 2.

**Training for Step t = 1** In the first step (lines 1-4), we simply train $\mathcal{S}_1, \mathcal{T}_1$ on full information i.e. on all features for total (chosen) $E$ epochs. At completion, we set the first sample-independent score $\bar{\mathbf{s}}_1$ (line 4) to be the mean of the sample-dependent scores $\tilde{s}$ across samples/batches. We found training solely on a sample-dependent score results in faster optimization.

**Training for Steps t = 2,...,T** The four phases require choosing the number of epochs for each phase: $1 <= E_1 < E_2 < E_3 < E$ for total number of epochs $E$, training proceeds as follows:

**Phase 1)** Initialize $\mathcal{S}_t$ and $\mathcal{T}_t$ from $\mathcal{S}_{t-1}$ and $\mathcal{T}_{t-1}$, $\bar{\mathbf{s}}_t$ to $\bar{\mathbf{s}}_{t-1}$, $\mathbf{m}_t$ to $\mathbf{m}_{t-1}$,e and $\alpha = \frac{1}{2}$ to balance learning a new score for this step and using information from the learnt score from step $t-1$. Run $E_1$ epochs to refine scores and task execution with $\alpha$ and $\mathbf{m}_t$ fixed.

**Phase 2)** Update the sample-independent score $\bar{\mathbf{s}}_t$ with the learnt score from phase 1 (line 11). Run $E_2 - E_1$ epochs progressively linearly modifying $\alpha$ (line 13), so training moves gradually from using sample-dependent scores to sample-independent.

**Phase 3)** Choose the $C_{t-1} - C_t$ lowest-scored features to remove (lines 16, 17). Run $E_3 - E_2$ epochs linearly modifying the mask for subsampling (line 19). This alters the $C_{t-1} - C_t$ elements of $\mathbf{m}_t$ corresponding to the lowest-scored features gradually to 0. Thus $\|\mathbf{m}_t\|_0 = C_{t-1}$ goes to $\|\mathbf{m}_t\|_0 = C_t$. Separating this phase from phase 2 increases the stability of the optimization, as modifying the mask and score simultaneously results in large gradients.

**Phase 4)** Train $\mathcal{T}_t$ for final refinement for $E - E_3$ epochs with the score weights fixed and features chosen. At completion return $\mathcal{T}_t, \mathbf{m}_t, \bar{\mathbf{s}}_t$.

**Implementation Details and Hyperparameters** TADRED's hyperparameters are fixed across experiments and different application areas. They are detailed in appendix A.

## 4 EXPERIMENTS AND RESULTS

This section demonstrates the benefits of TADRED in multiple scenarios, with example applications in qMRI and hyperspectral imaging. First, in table 1, we consider the standard experimental design task of model parameter estimation and outperform classical Fisher-matrix approaches. Within the new paradigm, we also show improvements over recent supervised feature selection approaches. We then show TADRED's efficacy in a 'model-free' experimental design scenario: reconstruction of a densely sampled data set from a sparse subset, where Fisher-matrix or recent Bayesian experimental design cannot operate and TADRED outperforms best published results in an MRI challenge in table 2. In figure 2 we consider a reconstruction task

---

**Algorithm 1** TADRED Forward & Backward Pass (FBP) in Step $t$

**Requires:**

Input and Target Data $X_{\bar{D}}, Y$, Mask $\mathbf{m}_t$
Scoring and Task Networks $\mathcal{S}_t, \mathcal{T}_t$, Loss $L$
Sample-independent Feature Score $\bar{\mathbf{s}}_t$
Mix Parameter $\alpha \in [0, 1]$, Feature Fill $X_{\bar{D}}^{\text{fill}}$

1: $\tilde{s} = \sigma(\mathcal{S}_t(X_{\bar{D}}))$
2: $\mathbf{s} = \alpha \odot \tilde{s} + (1 - \alpha) \odot \bar{\mathbf{s}}_t$ # Equation 4
3: $X_{D_t} = \mathbf{m}_t \odot X_{\bar{D}} + (\mathbf{1}_{\bar{C}} - \mathbf{m}_t) \odot X_{\bar{D}}^{\text{fill}}$
4: $\widehat{Y} = \mathcal{T}_t(\mathbf{s} \odot X_{D_t})$
5: Compute $L(\widehat{Y}, Y)$ and backpropagate

---

**Algorithm 2** TADRED Optimization

**Requires:**

Input and Target Data $X_{\bar{D}} \in \mathbb{R}^{n \times \bar{C}}, Y$
Loss $L$, Feature Fill $X_{\bar{D}}^{\text{fill}} \in \mathbb{R}^{\bar{C}}$
Feature Set Sizes $\bar{C} = C_1 > ... > C_T$
Training Steps $1 \le E_1 < E_2 < E_3 < E$
Initial Scoring and Task Networks $\mathcal{S}_1, \mathcal{T}_1$

1: $t \leftarrow 1$; $\mathbf{m}_1 \leftarrow \mathbf{1}_{\bar{C}}$; $\alpha \leftarrow 1$ # Step t = 1
2: **for** $e \leftarrow 1, ..., E$ **do**
3:     FBP()               # Algorithm 1
4: $\bar{\mathbf{s}}_1 \leftarrow$ mean of $\tilde{s}$ across data
5: **for** $t \leftarrow 2, ..., T$ **do**      # Steps $t \ge 2$
6:                            # Phase 1
7:     $\bar{\mathbf{s}}_t \leftarrow \bar{\mathbf{s}}_{t-1}$; $\alpha \leftarrow \frac{1}{2}$; $\mathbf{m}_t \leftarrow \mathbf{m}_{t-1}$
8:     **for** $e \leftarrow 1, ..., E_1$ **do**
9:         FBP()
10:     $\bar{\mathbf{s}} \leftarrow$ mean of $\tilde{s}$ on data    # Phase 2
11:     $\bar{\mathbf{s}}_t \leftarrow \frac{1}{2}(\bar{\mathbf{s}}_t + \bar{\mathbf{s}})$
12:     **for** $e \leftarrow E_1 + 1, ..., E_2$ **do**
13:         $\alpha \leftarrow \max\{\alpha - \frac{1}{2(E_2 - E_1)}, 0\}$
14:         FBP()
15:     # Indices that sort an array; Phase 3
16:     $R = \text{argsort}\{\bar{\mathbf{s}}_t[i] : \mathbf{m}_t[i] = 1\}$
17:     $D_t \leftarrow \{R[0], ..., R[C_{t-1} - C_t]\}$
18:     **for** $e \leftarrow E_2 + 1, ..., E_3$ **do**
19:         $\mathbf{m}_t \leftarrow \max\{\mathbf{m}_t - \frac{\mathbb{I}[i]_{i \in D_t}}{E_3 - E_2}, \mathbf{0}_{\bar{C}}\}$
20:         FBP()
21:     **for** $e \leftarrow E_3 + 1, ..., E$ **do** # Phase 4
22:         FBP()
23:     Cache $\mathcal{T}_t, \mathbf{m}_t, \bar{\mathbf{s}}_t$ for equation 3

---

to then estimate multiple clinically-relevant downstream metrics from model fitting - extending the traditional model-parameter estimation task to estimate multiple quantities. TADRED outperforms recent supervised feature selection techniques in this task that has immediate deployment potential. We then show the generalizability of TADRED by performing similar sets of experiments on hyperspectral images, outperforming both supervised feature selection baselines for earth remote sensing in table 3 and recent work in tissue oxygenation estimation in table 4. Tables 5, 6 show an ablation study and that TADRED is mostly robust to randomness in deep learning training.

Appendix C provides additional analysis. Appendix E provides details on experimental design in qMRI and hyperspectral imaging and how to implement our paradigm in real-world scenarios. Appendix F summarizes and visualizes the resultant densely-sampled data $X_{\bar{D}}$. Following standard practice in MR parameter estimation Alexander et al. (2019); Cercignani et al. (2018), and hyperspectral image filter design Waterhouse & Stoyanov (2022), data samples are individual pixels/voxels.

Table 1: Performance comparison of feature selection approaches for VERDICT-MRI designs: MSE $\times 10^2$ between estimated model parameters and ground truth for various $C$ and $\bar{C} = 220$.

|        | $C = 110$ | 55   | 28   | 14   |
|--------|-----------|------|------|------|
| Random | 1.54      | 2.24 | 3.25 | 6.10 |
| SSEFS  | 1.06      | 1.28 | 1.89 | 4.58 |
| FIRDL  | 2.22      | 2.14 | 3.09 | 4.05 |
| TADRED | **1.03**  | **1.18** | **1.80** | **2.64** |

Table 2: Performance comparison on MUDI: MSE between $\bar{C} = 1344$ reconstructed MRI channels/measurements and $\bar{C}$ ground-truth measurements for various $C$. PROSUB results from Blumberg et al. (2022) table 1.

|        | $C = 500$ | 250  | 100  | 50   |
|--------|-----------|------|------|------|
| PROSUB | 0.49      | 0.61 | 0.89 | 1.35 |
| TADRED | **0.22**  | **0.43** | **0.88** | **1.34** |
|        | $C = 40$  | 30   | 20   | 10   |
| PROSUB | 1.53      | 1.87 | 2.50 | 3.48 |
| TADRED | **1.52**  | **1.76** | **2.12** | **2.88** |

Figure 2: Downstream MRI metrics (see appendix F.3) estimated from the full set of channels/measurements on HCP data $\bar{C} = 288$, and $\bar{C}$ from $C = 18$ reconstructed measurements. Left: MSE for various metrics; Right: Qualitative comparison where arrows highlight closer agreement from TADRED's design with the gold standard than those from the best performing baseline.

|        | DTI | | | |
|--------|------|------|------|------|
|        | FA   | MD   | AD   | RD   |
| Random | 2.22 | 6.09 | 22.7 | 6.97 |
| SSEFS  | 2.86 | 12.9 | 31.2 | 14.9 |
| FIRDL  | 9.83 | 23.2 | 77.7 | 26.8 |
| TADRED | **1.29** | **2.55** | **13.4** | **2.60** |

|        | DKI | | | MSDKI | |
|--------|------|------|------|------|------|
|        | MK   | AK   | RK   | MSD  | MSK  |
| Random | 9.03 | 7.83 | 15.3 | 6.82 | 7.59 |
| SSEFS  | 12.0 | 9.26 | 20.3 | 8.96 | 8.17 |
| FIRDL  | 11.9 | 10.9 | 21.3 | 10.8 | 6.03 |
| TADRED | **7.67** | **6.73** | **13.9** | **6.37** | **4.94** |

**Baselines and Comparisons** We compare TADRED with standard model-based approaches such as the classical Fisher-matrix. Within the subsampling-task paradigm we use i) recent application-specific published results optimized by the respective authors; ii) state-of-the-art supervised feature selection approaches FIRDL, SSEFS (see section 2) and random selection then deep learning training (denoted by 'random') to mimic random baselines used in experimental design papers. Each feature selection approach conducts an extensive hyperparameter search, for fairness, the same number of evaluations are used for each feature subset $C$. All details are in appendix A. As this requires multiple training run (SSEFS in table 1 requires >400 runs), we examine the effect of the random seed on performance in table 6. We compare the computational costs of different approaches in appendix D.

**TADRED Outperforms Classical Experimental Design and Baselines in Model Parameter Estimation** A standard task in experimental design is selecting the design $D$ to maximize the precision of model parameters. We evaluate strategies for this using the VERDICT-MRI model which aids early detection and classification of prostate cancer Panagiotaki et al. (2015a). We sample parameters $\boldsymbol{\theta}_i$ for voxel $i = 1, ..., n$, from a biologically plausible range, add synthetic noise representative of clinical qMRI, and the task is to estimate $Y = \{\boldsymbol{\theta}_1, ..., \boldsymbol{\theta}_n\}$ with performance metric MSE. The first baseline Panagiotaki et al. (2015b) uses classical Fisher-matrix experimental design (see section 2), to compute the design $D$ with $C = 20$. The design produces a root-mean square error of $15.0 \times 10^{-2}$ in this experiment. The TADRED design with $C = 20$ has corresponding error of $2.04 \times 10^{-2}$. The supervised feature selection approaches in the new paradigm use a densely-sampled design $\bar{D}$, where $\bar{C} = 220$ from Panagiotaki et al. (2015a) and use deep learning to estimate $\boldsymbol{\theta}_i$. Appendix F.1 documents all designs, models, and data. Table 1 shows TADRED outperforms the feature selection baselines where $C = \frac{\bar{C}}{2}, \frac{\bar{C}}{4}, \frac{\bar{C}}{8}, \frac{\bar{C}}{16}$. Thus it can better estimate parameters shown to reduce unnecessary biopsies Singh et al. (2022) in shorter scan times, spurring wider deployment in clinical settings. Similar results on the well-known NODDI model are in appendix B.

**Best Performance on qMRI Challenge Data** The Multi-Diffusion Challenge Pizzolato et al. (2020) aimed to identify an informative subset of data, from which to reconstruct the original full dataset $X_{\bar{D}}$ (i.e. $Y = X_{\bar{D}}$) which had $\bar{C} = 1344$ measurements. This task provides a generic challenge that tests the ability of an experimental design or supervised feature selection algorithm to identify a

Table 3: Performance comparison of feature selection approaches for remote sensing AVIRIS hyperspectral data, MSE between $\bar{C} = 220$ reconstructed and $\bar{C}$ ground-truth measurements.

| | $C = 110$ | 55 | 28 | 14 |
|---|---|---|---|---|
| Random | 1.81 | 2.60 | 3.99 | 8.27 |
| SSEFS | 2.03 | 4.49 | 5.77 | 10.8 |
| FIRDL | 8.10 | 9.87 | 10.6 | 10.3 |
| TADRED | **0.87** | **1.82** | **2.84** | **5.80** |

Table 4: Performance comparison RMSE $\times 10^2$, estimating abundance of $HbO_2, Hb$ (top), $SO_2$ (bottom). Experimental settings and baseline from Waterhouse & Stoyanov (2022) figure 5.

| | $C = 6$ | 5 | 4 | 3 |
|---|---|---|---|---|
| Baseline | 4.54 | 4.91 | 5.33 | 6.17 |
| TADRED | **2.80** | **2.89** | **3.23** | **4.36** |
| | $C = 6$ | 5 | 4 | 3 |
| Baseline | 4.45 | 4.43 | 5.10 | 6.36 |
| TADRED | **2.76** | **2.94** | **3.46** | **5.64** |

subset with maximal information content. As discussed in section 2, neither classical Fisher matrix nor Bayesian experimental design approaches can perform this task. Data are brain scans of five human subjects, which were acquired from a state-of-the-art technique that acquires multiple MRI modalities simultaneously in a high-dimensional space where $\mathbf{d}^i \in \mathbb{R}^6$. Thus experimental design is important, as sampling in a time budget realistic in clinical settings is difficult Slator et al. (2021). The first experiment follows Blumberg et al. (2022) which has the best performance on the data and table 2 shows TADRED outperforms this approach. We also show TADRED outperforms the supervised feature selection baselines in appendix B. All details are in appendix F.2.

**Surpassing the Baselines in Estimation of Multiple Downstream Metrics** DTI, DKI, and MSDKI Basser et al. (1994); Jensen & Helpern (2010); Henriques (2018) are widely-used qMRI methods. They quantify tissue microstructure and show promise for extracting imaging biomarkers for many medical applications, such as mild brain trauma, epilepsy, stroke, and Alzheimer's disease Jensen & Helpern (2010); Ranzenberger & Snyder (2022); Tae et al. (2018). Reducing acquisition requirements (picking a small $C$) whilst obtaining more accurate quantification will enable their usage in a wider range of clinical application areas. We use publicly available, rich, high-resolution HCP data with $\bar{C} = 288$ measurements from six human subjects, corresponding to $\approx 30$ minute scan times in the clinic – too long for general deployment. The task is to subsample sizes $C = \frac{\bar{C}}{8}, \frac{\bar{C}}{16}$ then reconstruct the data, where the models are then fitted using standard techniques. Further details on the models, data, and model fitting techniques are in appendix F.3. Quantitative and qualitative results are in figure 2 and appendix B and show TADRED outperforms the baselines on 17/18 comparisons on clinically useful downstream metrics. Furthermore, the downstream metrics produced by TADRED are visually closer to the gold standard than those from the best baseline, potentially enhancing the diagnosis of aberrations in tissue microstructure.

**Outperforming Baselines in Reconstructing Remote Sensing Ground Images** The JPL's Airborne Visible / Infrared Imaging Spectrometer (AVIRIS) Thompson et al. (2017) remotely senses elements of the Earth's atmosphere and surface from aeroplanes, and has been used to examine the effect and rehabilitation of forests affected by large wildfires. Purdue University Agronomy Department obtained AVIRIS data to support soils research and we use this publicly available 'Indian Pine' data Baumgardner et al. (2015), obtained from two flights - which acquired ground images from $\overline{C} = 220$ different wavelengths. Details are in appendix F.4. This experiment follows experiment in table 2 and examines a sampling-reconstruction task where we investigate if we can obtain the same quality data with fewer wavelengths – which would in practice require fewer sensors. Table 3 shows TADRED outperforms the supervised feature selection baselines with subsample sizes $C = \frac{\bar{C}}{2}, \frac{\bar{C}}{4}, \frac{\bar{C}}{8}, \frac{\bar{C}}{16}$. These improvements demonstrate the potential for using fewer filters in AVIRIS. In the development of next-generation airborne hyperspectral devices, TADRED may be used to choose the filters. Further results are in appendix B, promising additional applications are outlined in appendix E.

**Improving the Estimation of Oxygen Saturation** This experiment follows Waterhouse & Stoyanov (2022). Tissue oxygen saturation levels provide information regarding chemical and heat burns, along with the likelihood of healing. However, techniques such as spectrophotometers and pulse oximetry do not provide the spatial resolution to observe differences in blood saturation in neighboring tissue. Hyperspectral imaging is a non-invasive and real-time alternative to improve oxygenation estimation, yet application-specific spectral band selection is required to reduce the high cost of imaging sensors, allowing widespread clinical adoption. To address this, Waterhouse & Stoyanov (2022) adapted the model in Can & Ülgen (2019) for simulations and the objective is to estimate the pixel-wise abundance of oxyhemoglobin $HbO_2$ and deoxyhemoglobin $Hb$; and oxygen saturation $SO_2$. Design

Table 5: Ablation study on TADRED's components.

| | $C = 110$ | 55 | 28 | 14 |
|---|---|---|---|---|
| w/o Scoring Network $S$ | 7.23 | 10.7 | 11.5 | 11.5 |
| w/o iterative subsampling | **1.03** | 1.19 | 1.83 | 2.80 |
| TADRED | **1.03** | **1.18** | **1.80** | **2.64** |

Table 6: Standard deviation of performance $\times 10^2$ across 10 random seeds settings.

| | $C = 110$ | 55 | 28 | 14 |
|---|---|---|---|---|
| Random | 0.11 | 0.23 | 0.37 | 0.76 |
| SSEFS | 0.01 | 0.02 | 0.05 | 0.18 |
| FIRDL | 0.44 | 0.34 | 0.37 | 0.44 |
| TADRED | 0.01 | 0.02 | 0.01 | 0.12 |

elements $\mathbf{d}^i \in \bar{D}$ are chosen from 4 filters of different widths applied to 87 wavelengths (center), producing $\bar{C} = 348$ measurements. Table 4 shows that TADRED produces directly outperforms all approaches and results published and optimized in Waterhouse & Stoyanov (2022) for estimating the abundance of $HbO_2, Hb, SO_2$; for feature sizes $C = 6, 5, 4, 3$. This suggests that using TADRED during the development of clinically-viable hyperspectral devices may be beneficial to reduce costs.

**Component Analysis and the Effect of Randomness** We use the experimental settings in table 1. Table 5 examines the impact of removing TADRED's components on performance. First it considers TADRED without iteratively removing features in the optimization procedure, fixing $t = 2$ and $C_1, C_2 = \bar{C}, C$, showing that iterative subsampling has better performance than subsampling all features one iteration. As the feature scoring is a key element of TADRED, we also show that removing the scoring network $\mathcal{S}$, whilst still learning a score, results in extremely poor performance, as training is destabilized when progressively setting the score from sample-dependent to sample-independent (recall equation 4). Table 6 shows how changing the random seed affects network initialization and data shuffling impacts performance; TADRED performs favorably compared to alternative approaches, and TADRED is mostly robust to the randomness inherent in deep learning.

## 5 DISCUSSION

This paper proposes TADRED, a feature selection algorithm that enables a new subsampling paradigm for experimental design particularly in multi-channel imaging applications. We demonstrate substantial performance benefits over standard Fisher-matrix approaches at the heart of widely used quantitative MRI techniques, as well as strong potential in multiple hyperspectral-imaging applications. "Standard" data sets for testing TADRED do not exist, as its new paradigm is largely unexplored, but in the few available examples (dataset used in table 2 and hyperspectral datasets in tables 3, 4) TADRED strongly outperforms existing algorithms, even on datasets for which those baselines were specifically designed, and without problem-specific hyperparameter tuning.

TADRED combines the dual selection/task network training strategy in state-of-the-art feature selection algorithms (SSEFS and FIRDL) with an RFE framework better suited to identifying complementary subsets among many informative candidate features. Thus, TADRED outperforms SSEFS and FIRDL on the imaging experimental design problems we consider. In fact, random supervised feature selection often outperforms SSEFS when there are no informative/correlated feature subsets to identify and FIRDL's complex optimization procedure is often not beneficial when there is no such subset to identify and it underperforms simpler approaches. On the other hand, TADRED is likely to underperform SSEFS and FIRDL on typical applications in supervised feature selection where small sets of discriminative features reside among many uninformative features. One possible limitation is that TADRED's iterative subsampling in the paradigm of RFE and backward selection decreases the upper bound on performance as the optimal feature sets for sizes $C_t, C_{t-1}$ may not be nested. Future work will consider alternative strategies. This iterative subsampling also increases computational time compared to random supervised feature selection. However, appendix D shows TADRED's computational time is comparable to SSEFS and FIRDL. Here, we consider all image channels to have equal cost, but in practice some measurements/channels may be more expensive than others; TADRED's formulation adapts naturally to more complex cost functions on the experimental design. Also here we consider only tasks that treat each image pixel/voxel independently which is typical in quantitative imaging Alexander et al. (2019); Cercignani et al. (2018), so use only fully-connected networks (as the baselines), again TADRED's formulation adapts naturally to use e.g. a CNN for $\mathcal{T}$. TADRED has further applications to other imaging problems e.g autofocus for specialized equipment Lightley et al. (2022) and potentially beyond imaging to e.g. studies of cell populations Sinkoe & Hahn (2017).

## REPRODUCIBILITY STATEMENT

We provide the code: Code Link, which contains the entire source code for our algorithm TADRED. The code also contains the script to create simulations used in tables 1, 7 and downloading and preprocessing the data in for results presented in tables 2, 3 and figure 2. Further details on all data and preprocessing are in appendix F. We also provide detailed information on the implementation of TADRED and the baselines in appendix A.

## ACKNOWLEDGEMENTS

HPC: Tristan Clark, James O'Connor, Edward Martin; Ahmed Abdelkarim, Daniel Beechey, George Blumberg, Răzvan Căramaluău, Amy Chapman, Alice Cheng, Luca Franceschi, G-Research (for a previous grant), Fredrik Helltröm, Jessica Hoang, Chen Jin, Jean Kaddour, Marcus Keil, Johannes Kirschner, Marcela Konanova, Eve Levy and Michael Salvato, Hongxiang Lin, Nina Montaña-Brown, Luca Morreale, MUDI Organizers, Raymond Ojinnaka, Gabriel Oon, Brooks Paige, David Pérez-Suárez, Stefan Piatek, Reviewers, Oliver Slumbers, Dennis Soemers, Danail Stoyanov, Shinichi Tamura, Dale Waterhouse, Tom Young, An Zhao, Yukun Zhou. Funding: EPSRC grants M020533 R006032 R014019, Microsoft scholarship, NIHR UCLH Biomedical Research Centre, Research Initiation Project of Zhejiang Lab (No.2021ND0PI02). Data were provided [in part] by the Human Connectome Project, MGH-USC Consortium (Principal Investigators: Bruce R. Rosen, Arthur W. Toga and Van Wedeen; U01MH093765) funded by the NIH Blueprint Initiative for Neuroscience Research grant; the National Institutes of Health grant P41EB015896; and the Instrumentation Grants S10RR023043, 1S10RR023401, 1S10RR019307.

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

## APPENDIX STRUCTURE

The appendices are structured as follows:

1. Appendix A provides comprehensive details on all approaches utilized in this paper, including the specific hyperparameters employed.
2. Appendix B are supplementary experimental results.
3. Appendix C offers further experimental analysis of our method, TADRED.
4. Appendix D compares the computational cost of all the approaches used in this paper and outlines the computational resources employed.
5. Appendix E is a comprehensive description of prior work related to our problem.
6. Appendix F details all data for each experiment, along with specifics of each task.

## A KEY APPROACHES, HYPERPARAMETERS, AND SETTINGS

This section describes the different supervised feature selection approaches used in this paper and details the choice of parameter settings within each.

### GENERAL EXPERIMENTAL SETTINGS

For every experiment comparing TADRED with baselines, we split the data into training, validation/development, and test sets. This is described in detail in section F. Following Lee et al. (2022) (paper of SSEFS), we conducted an extensive hyperparameter search for each approach (using the validation set), for different experimental settings and subsample values $C$. This is described in detail below for every approach. For fairness, we have the same number of evaluations on the validation set for each different feature set size $C$, i.e. number of trials for model selection. The best model was then applied to the test set and we reported the performance.

Other general hyperparameters are: batch size $1500$, learning rate $10^{-4}$ ($10^{-5}$ for the experiment in figure 2), ADAM optimizer, and default network weight initialization. The default option for early stopping used 20 epochs for patience (i.e training stops if validation performance does not improve in 20 epochs).

### TADRED - TASK-DRIVEN EXPERIMENTAL DESIGN FOR MULTI-CHANNEL IMAGING

This subsection details the hyperparameters for the method we present in this paper: TADRED for TAsk-DRiven Experimental Design in imaging, as outlined in section 3. Figure 3 provides a graphical representation of TADRED's structure and computational graph.

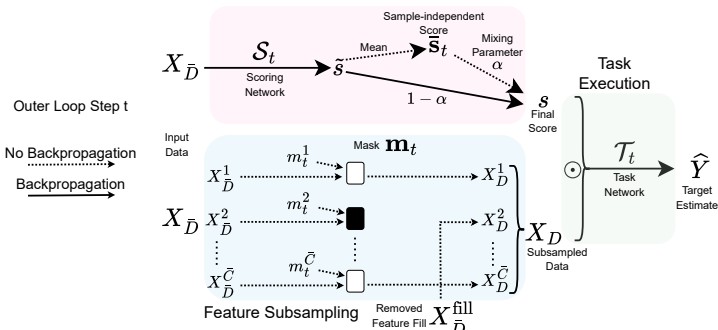

Figure 3: TADRED's structure. During training TADRED concurrently performs feature scoring, feature subsampling, and task execution. During training we progressively set the score to be sample-independent by setting $\alpha$ to 1. We score features with $\bar{\mathbf{s}}_t \in \mathbb{R}^{\bar{C}}$ and remove features with low score by setting corresponding values of the mask to 0, in this example we removed feature 2.

We conducted a brief search for TADRED-specific hyperparameters and fixed these hyperparameters across all experiments. We set the numbers of epochs in the four-phase inner loop training procedure as $E_1 = 25, E_2 = E_1 + 10, E_3 = E_2 + 10$. Following other baselines, we do not fix the total number of training epochs $E$, but keep training beyond $e = E_3$ in algorithm 2 phase 4 until early stopping criteria (on the validation set) are met.

For fairness, when comparing TADRED with other supervised feature selection baselines, we chose a simple set of hyperparameters for the feature set sizes $\{C_t\}_{t=1}^T$ and number of outer loop steps $T$. Here we fixed $T = 5$ and $C_1, C_2, C_3, C_4, C_5 = \bar{C}, \frac{\bar{C}}{2}, \frac{\bar{C}}{4}, \frac{\bar{C}}{8}, \frac{\bar{C}}{16}$. When comparing TADRED against two recent application-specific published results optimized by the authors of Blumberg et al. (2022); Waterhouse & Stoyanov (2022), we followed standard practice and performed a brief hyperparameter search on the validation set. We used $C_1, ..., C_9 = \{1344, 500, 250, 100, 50, 40, 30, 20, 10\}, T = 9$ in table 2 and $C_1, ..., C_{19} = [348] + [250::45::-50] + [45::8::-5] + [8,6,5,4,3,2], T = 19$ (notation is [start::stop::step] ) in table 4.

We perform a grid search to find the optimal network architecture hyperparameters for each task. The Scoring Network $\mathcal{S}$ and Task Network $\mathcal{T}$ have the same number of hidden layers $\in \{1, 2, 3\}$, number of units $\in \{30, 100, 300, 1000, 3000\}$, and for each combination we obtain task performance on the feature set sizes $C_1 > C_2 > ... > C_T$. The best performing network on the validation set is deployed on the test data.

RANDOM SUPERVISED FEATURE SELECTION

This baseline is inspired by the random design baselines used in experimental design papers e.g. Foster et al. (2021); Ivanova et al. (2021). For a particular design size, $C$, we repeat the following process: i) randomly select $C$ features/channels; ii) perform grid search on the task network (mapping subsampled data $X_{\bar{D}}$ to target $Y$), with number of hidden layers $\in \{1, 2, 3\}$, number of units $\in \{30, 100, 300, 1000, 3000\}$; iii) train until early stopping criteria specified on the validation set are met; iv) evaluate the best trained model on the test set.

SELF-SUPERVISION ENHANCED FEATURE SELECTION WITH CORRELATED GATES (SSEFS) LEE ET AL. (2022)

This approach has a lengthy hyperparameter search detailed in Appendix B of Lee et al. (2022), which consists of a three-phase procedure and four neural networks. Note that this required multiple training steps e.g. obtaining results for in table 1 requires >400 runs. SSEFS exploits task-performance, self-supervision, additional unlabeled data, and correlated feature subsets. It scores the features then subsequently trains a task-based network (analogous to $\mathcal{T}$ in TADRED) on subsampled data. We use the official repository Lee (2022) and verified our implementation by replicating results in the

paper Lee et al. (2022). The full optimization procedure follows Lee et al. (2022) and is split into i) self-supervision phase, ii) supervision phase, iii) training on selected features only.

The self-supervision phase finds the optimal encoder network hyperparameters. We follow Appendix B of Lee et al. (2022) and perform grid search. Similar to other approaches in this paper, we consider the encoder network, feature vector estimator network, gate vector estimator network, all have same number of hidden layers $\in \{1, 2, 3\}$ number of units, including hidden dimension $\in \{30, 100, 300, 1000, 3000\}$. Directly following Lee et al. (2022) table S.1, other hyperparameters $\alpha \in \{0.01, 0.1, 1.0, 10, 100\}, \pi \in \{0.2, 0.4, 0.6, 0.8\}$. The self-supervisory dataset is input data $X_{\bar{D}}$. On the best validation performance (with early stopping), this returns a trained encoder network, cached for the supervision phase.

The supervision phase scores the features. The pretrained encoder is loaded from the previous phase. We then perform grid search, where the predictor network has number of hidden layers $\in \{1, 2, 3\}$, number of units $\in \{30, 100, 300, 1000, 3000\}$, following Lee et al. (2022) table S.1 $\beta \in \{0.01, 0.1, 1.0, 10, 100\}$. On the best validation performance with early stopping, the process returns a score for all features.

The final phase is repeated for a different number of subset sizes $C$. We extract the $C$ highest scored features from the previous phase and perform grid search on the task network (mapping subsampled data $X_{\bar{D}}$ to target $Y$), with number of hidden layers $\in \{1, 2, 3\}$, number of units $\in \{30, 100, 300, 1000, 3000\}$. Training is until early stopping on the validation set. The best trained model is evaluated on the test set.

FEATURE IMPORTANCE RANKING FOR DEEP LEARNING (FIRDL) WOJTAS & CHEN (2020)

This approach has a three-stage procedure detailed in Appendix D of Wojtas & Chen (2020) and uses two neural networks. One of the networks scores the masks (analogous to mask **m** in TADRED), and the other trains a task network to perform a task on the subsampled data (analogous to $\mathcal{T}$ in TADRED). We use the official repository Wojtas (2021) and verified our implementation by replicating results in the paper Wojtas & Chen (2020). The following process is repeated for different feature subset sizes $C$.

We perform a grid search to find the optimal hyperparameters. The operator network (analogous to task network $\mathcal{T}$ in this paper) and the selector network have the same number of hidden layers $\in \{1, 2, 3\}$, number of units $\in \{30, 100, 300, 1000, 3000\}, s_p = 5, E_1 = 15000$. The joint training uses early stopping on the validation set, and returns an optimal feature set size, of size $C$ and a trained operator network. The best performing operator network on the validation set is deployed on the test data.

# B  ADDITIONAL RESULTS

This section provides additional results supporting the experiments presented in the main paper.

Table 7 contains results that repeat the experiment in table 1 using the NODDI model Zhang et al. (2012) instead of VERDICT. The first baseline uses the classical Fisher-matrix experimental design Alexander (2008) to compute the design $D$ from Zhang et al. (2012) where $C = 99$. For the supervised feature selection approaches we use densely-sampled designs $\bar{D}$ where $\bar{C} = 3612$ Ferizi et al. (2017). Similar to results in table 1, table 7 shows TADRED outperforms classical experimental design with $C$ set to 99 following the classical approach used in current practice. In addition, TADRED outperforms the supervised feature selection baselines where $C = \frac{\bar{C}}{2}, \frac{\bar{C}}{4}, \frac{\bar{C}}{8}, \frac{\bar{C}}{16}$. The optimized designs enable us to estimate the widely used NODDI parameters in shorter scan times opening the potential for a wider range of clinical applications. All information on designs, models and data is in section F.1.

Table 8 shows extra results within the experiment documented in figure 2. We consider an additional feature set subsample size $C = 36$ which extends the results in figure 2 and show TADRED outperforms the baselines on 17/18 comparisons on clinically useful downstream metrics. This is beneficial as pressure for time in clinical MRI protocols is intense as many different MR contrasts are informative, but patient-time in the scanner is limited. Therefore shorter acquisition protocols for

Table 7: MSE $\times 10^2$ between estimated NODDI model parameters and ground truth parameters used to simulate the data. Classical experimental design is from Zhang et al. (2012) and uses a Fisher-matrix approach. Additional results for table 1.

| Classic Experimental Design $C = 99$ | | | 8.00 |
|---|---|---|---|
| TADRED $C = 99, \bar{C} = 3612$ | | | **4.51** |

| $\bar{C} = 3612$ | $C = 1806$ | 903 | 452 | 226 |
|---|---|---|---|---|
| Random | 2.99 | 3.34 | 3.88 | 4.31 |
| SSEFS | 2.95 | 3.39 | 3.73 | 4.39 |
| FIRDL | 4.21 | 4.61 | 4.96 | 5.14 |
| TADRED | **2.59** | **2.92** | **3.33** | **3.85** |

Table 8: MSE for downstream MRI metrics (see appendix F.3) estimated from the full set of measurements on HCP data $\bar{C} = 288$, and $\bar{C}$ from $C = 36$ reconstructed measurements. Additional results to figure 2.

| | DTI | | | |
|---|---|---|---|---|
| | FA | MD | AD | RD |
| Random | 1.17 | 3.13 | 9.3 | 3.75 |
| SSEFS | 5.96 | 10.4 | 43.7 | 14.2 |
| FIRDL | 10.4 | 68.8 | 106 | 81.4 |
| TADRED | **0.94** | **1.73** | **8.15** | **1.50** |

| | DKI | | | MSDKI | |
|---|---|---|---|---|---|
| | MK | AK | RK | MSD | MSK |
| Random | 6.37 | 6.28 | 13.3 | **2.50** | 4.15 |
| SSEFS | 8.74 | 7.27 | 16.9 | 3.87 | 10.8 |
| FIRDL | 9.39 | 10.3 | 18.6 | 11.1 | 8.82 |
| TADRED | **6.15** | **5.92** | **12.6** | 2.75 | **3.78** |

Table 9: MSE between $\bar{C} = 1344$ reconstructed measurements and $\bar{C}$ ground-truth measurements on Multi-Diffusion challenge subjects. Additional results to table 2.

| | $C = 500$ | 250 | 100 | 50 |
|---|---|---|---|---|
| Random | 0.93 | 1.41 | 2.12 | 5.23 |
| SSEFS | 0.63 | 0.86 | 1.24 | 1.61 |
| FIRDL | 1.67 | 1.72 | 2.17 | 2.34 |
| TADRED | **0.21** | **0.44** | **0.94** | **1.34** |

Table 10: Performance comparison of feature selection approaches for remote sensing AVIRIS hyperspectral data (east-to-west flight of Indian Pine), MSE between $\bar{C} = 220$ reconstructed and $\bar{C}$ ground-truth measurements. Additional results to table 3.

| | $C = 110$ | 55 | 28 | 14 |
|---|---|---|---|---|
| Random | 1.22 | 1.76 | 2.91 | 5.61 |
| SSEFS | 1.36 | 3.11 | 3.77 | 7.61 |
| FIRDL | 6.34 | 6.68 | 7.16 | 7.78 |
| TADRED | **0.60** | **1.42** | **2.33** | **4.49** |

these widely informative downstream metrics (parametric maps) enables their exploitation in a wider range of clinical studies and applications.

Table 9 shows additional results on the MUDI data in table 2. This experiment compares TADRED with the supervised feature selection baselines following settings in the original MUDI challenge. Evaluation uses the MSE metric as in the original challenge. Further details are in appendix F.2.

Table 10 shows additional results on the AVIRIS data presented in table 3. Here, we only use data from the north-to-south flight. Improvements of TADRED over the supervised feature selection baselines are similar to that in table 3.

## C  FURTHER ANALYSIS

### C.1  ANALYZING THE EFFECT ON RANDOMNESS ON FEATURE SET CHOSEN

Table 11 examines how the changing the random seed that affects network initialization and data shuffling, impacts the feature set chosen. Results show TADRED performs favorably compared to alternative approaches and mostly chooses the same features

### C.2  EVALUATION OF THE CHOICE OF FEATURE FILL $X_{\bar{D}}^{\text{FILL}}$

Table 12 examines the effect of varying the values $X_{\bar{D}}^{\text{fill}}$ that fill the unsubsampled features. FIRDL used zeros for its equivalent of $X_{\bar{D}}^{\text{fill}}$ and SSEFS used the data mean (per channel/feature). Results show even if we set the values of $X_{\bar{D}}^{\text{fill}}$ to that of the baselines, TADRED has large improvements over the baselines.

### C.3  HOW DOES THE SIZE OF THE DENSELY-SAMPLED DESIGN AFFECT PERFORMANCE?

Table 11: Mean Jaccard Index between chosen measurements, across 10 random seeds, experimental settings in table 1 VERDICT simulations.

|        | $C = 110$ | 55   | 28   | 14   |
|--------|-----------|------|------|------|
| Random | 32.8      | 15.2 | 6.82 | 2.87 |
| SSEFS  | 81.2      | 71.4 | 62.2 | 75.9 |
| FIRDL  | 34.3      | 18.1 | 48.8 | 41.0 |
| TADRED | 74.3      | 82.1 | 84.6 | 59.1 |

Table 12: Comparison of the choice of $X_{\bar{D}}^{\text{fill}}$. Experimental settings follow table 1.

| | $X_{\bar{D}}^{\text{fill}}$ | $C = 110$ | 55 | 28 | 14 |
|--------|-------------|-----------|------|------|------|
| SSEFS  | data mean   | 1.06      | 1.28 | 1.89 | 4.58 |
| FIRDL  | zeros       | 2.22      | 2.14 | 3.09 | 4.05 |
| TADRED | data median | 1.03      | 1.19 | 1.80 | 2.55 |
| TADRED | data mean   | 1.03      | 1.20 | 1.79 | 2.80 |
| TADRED | zeros       | 1.03      | 1.19 | 1.79 | 2.51 |

We examine how varying the size of densely-sampled design $\bar{D}$ (used to create $X_{\bar{D}}$) affects performance. Across 10 random seeds, we randomly sample the design from Panagiotaki et al. (2015b) to create a custom $\bar{D}$ with $\bar{C}$ elements. Training is on fixed network sizes for a single subsampling rate $C_2 = C = 14$. We use 10% of the training data within the experimental settings of table 1. Results are in figure 4 and exemplify typical behavior that while performance is reasonably stable for large $\bar{C}$ a phase change occurs as $\bar{C}$ nears $C$ and performance decreases rapidly, as the set of samples to choose from becomes too sparse.

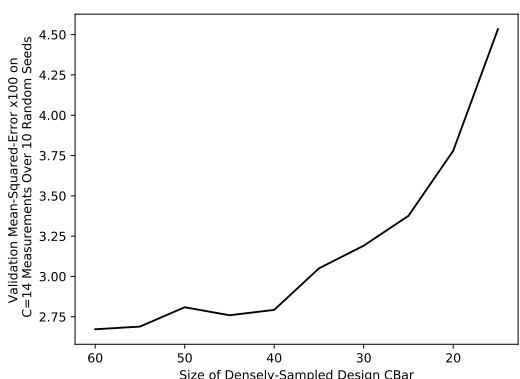

Figure 4: Analyzing the performance on different densely-sampled designs $\bar{D}$ where $|\bar{D}| = \bar{C}$ and $C = 14$. Settings follow table 1.

## C.4 TADRED
### VARIANT WITH RANDOM SELECTION

We tested a modified training procedure that works in the 'same manner' as the original implementation of TADRED whilst the scores chosen are random. Across different modifications, in settings of table 5, results (MSE) are more than 10% worse, even more than 'w/o iterative subsampling'. Thus, although the 'gradual aspect' of TADRED's training procedure improves performance (in fact line 2 in the ablation study table 5 already demonstrates this with 'less gradual' scenario without iterative subsampling decreases performance), the learning of the scoring network is working as intended and further improves results.

## D COMPUTATIONAL COST OF DIFFERENT APPROACHES AND INFRASTRUCTURE

It is difficult to compare the computational cost of TADRED against SSEFS, FIRDL. The official implementations, described in appendix A use different machine learning frameworks and all use customized early stopping. In particular, SSEFS has a three stage procedure (first two are large hyperparameter searches) which are completed consecutively; TADRED does not require this. TADRED and FIRDL train on two distinct networks, whilst SSEFS uses four, as such, training costs are somewhat comparable if network sizes are taken to be the same. Practical requirements in all cases were reasonable and training for all methods for each experiment were performed within 24 hours. As an example, we compare the time to run the various methods for the results in table 6 per $C$ with the network sizes fixed and no hyperparameter search over different network sizes. Training speeds are: Random supervised feature selection (baseline) 505s; SSEFS (baseline) 1934s (using only run a single run per seed for the first two stages; as previously noted, for other results in the paper, this is much slower as the method proposes a computationally expensive sequential hyperparameter search); FIRDL (baseline) 1756s; TADRED (the new approach) 1988s. The random supervised feature selection baseline is by far the most computationally economical, as we expect, because it uses no iterative search. TADRED's computational cost is similar to the two state-of-the-art supervised feature selection baselines, SSEFS and FIRDL.

Exploratory analysis and development was conducted on a mid-range (as of 2023) machine with a AMD Ryzen Threadripper 2950X CPU and single Titan V GPU. All experimental results reported in this paper were computed on low-to-mid range (as of 2023) graphical processing units (GPU): GTX 1080 Ti, Titan Xp, Titan X, Titan V, RTX 2080 Ti. We ran jobs on a high-performance computing cluster shared with other users, allowing multiple jobs to run in parallel.

# E  EXTENDED RELATED WORK

This section provides further information to section 2, detailing previous work related to our problem.

**Classical and Other Recent Supervised Feature Selection Approaches** Supervised feature selection approaches are either i) 'filter methods', which select features using some proxy metric independent of the final task, ii) 'wrapper methods', which use the task to evaluate feature set performance, or iii) 'embedded methods', which couple the feature selection with the task training. Embedded methods FIRDL Wojtas & Chen (2020), SSEFS Lee et al. (2022) are state-of-the-art outperforming classical approaches e.g recursive feature elimination (RFE)-original Guyon et al. (2002), BAHSIC Song et al. (2007; 2012), mRMR Peng et al. (2005), CCM Chen et al. (2017), RF Breiman (2001), DFS Li et al. (2016), LASSO Tibshirani (1996), L-Score He et al. (2005) and recent deep learning-based CE Abid et al. (2019), STG Yamada et al. (2020), DUFS Lindenbaum et al. (2021). More recent approaches extend the supervised feature selection paradigm to limit false discovery rate Hansen et al. (2022), few-shot learning Kumagai et al. (2022), discovering groups of predictive features Imrie et al. (2022), the unsupervised setting Sokar et al. (2022), few-sample classification problems Cohen et al. (2023), dynamic feature selection Covert et al. (2023), for federated learning Castiglia et al. (2023), for identifying high-dimensional causal features Quinzan et al. (2023). They are not designed for the standard regression-based supervised feature selection problem considered in this paper.

**Other Recent Experimental Design Approaches** Techniques for experimental design have been developed for causal modeling Tigas et al. (2022); Zhang et al. (2022); Teshnizi et al. (2020), linear models Fontaine et al. (2021); Mutny & Krause (2022), online learning Arbour et al. (2022), active learning Kaddour et al. (2020), drug discovery Mehrjou et al. (2022), reinforcement learning Mehta et al. (2022), A/B testing Nandy et al. (2021), panel-data settings Doudchenko et al. (2021), bandit problems Camilleri et al. (2021), balancing competing objectives with uncertainty Malkomes et al. (2021), temporal treatment and control Glynn et al. (2020), causal discovery when interventions can be costly or risky Tigas et al. (2023), designing pricing experiments Simchi-Levi & Wang (2023), contextual optimization for Bayesian experimental design Ivanova et al. (2023), genomics Lyle et al. (2023), treatment effects in large randomized trials Connolly et al. (2023), learning causal models with Bayesian approaches Annadani et al. (2023). These are not applicable to the problem setting we consider. Approaches Zheng et al. (2020); Kleinegesse & Gutmann (2020); Jiang et al. (2020) are older sequential experimental design approaches, whilst Zaballa & Hui (2023) is contemporary to this work – they face the same issues as Blau et al. (2022); Foster et al. (2021); Ivanova et al. (2021) (discussed in section 2) – which focus on estimating model parameters and are mostly demonstrated in small-scale problems which do not scale up to the high dimensional problems we face in experimental design for image-channel selection.

**Experimental Design in qMRI** In qMRI the design $D$ is known as an 'acquisition scheme'. One standard task is 'parameter mapping', first estimating biologically-informative model parameters by voxel-wise model fitting, to then obtain downstream metrics Alexander et al. (2019). This provides information that is not visible directly from the images, such as microstructural properties of tissue. However, acquisition time (corresponding to $C = |D|$) is limited by factors of cost and the ability of (often sick) subjects to remain motionless in the noisy and claustrophobic environment of the scanner. Thus experimental design can be crucial to support the most accurate image-driven diagnosis, prognosis, or treatment choices. Many clinical scenarios use $D$ based on intuition loosely guided by understanding of the physical systems under examination, but this can lead to highly suboptimal designs particularly for complex models. However, some studies optimize the design using the Fisher information matrix, e.g. Alexander (2008); Cercignani & Alexander (2006).

Lengthy MRI acquisitions corresponding to $\bar{D}$ to enable our new experimental design paradigm are made easily on a few subjects, but are not feasible in routine patient imaging. However, such lengthy acquisitions are often made in the design phase of quantitative imaging techniques, e.g. as in Ferizi et al. (2017). We note also that several distinct experiment design problems arise in MRI.

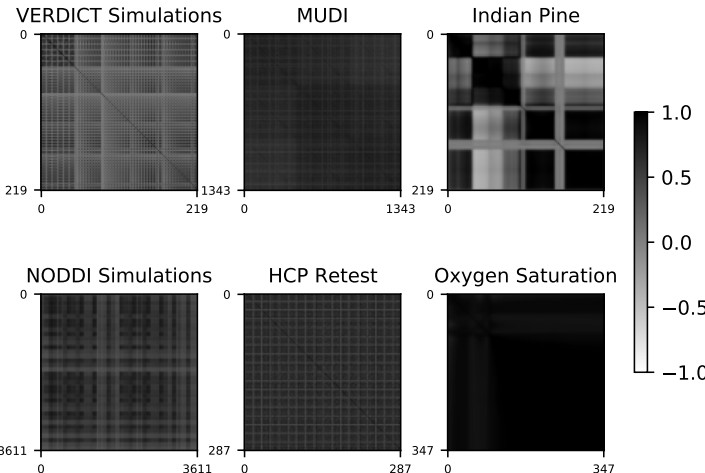

Figure 5: Correlation coefficient between the measurements/features/channels of the data.

Here we focus on estimating per-voxel parameter values, but others e.g. Zbontar et al. (2018); Knoll et al. (2020); Muckley et al. (2021); Fabian et al. (2022); Yaman et al. (2022), focus on how best to subsample the k-space. Our approach is complementary to and may be combined with those; they expedite the acquisition of each individual channel, we identify a compact/economical set of channels.

**Experimental Design in Hyperspectral Imaging** Hyperspectral imaging (a.k.a. imaging spectroscopy) obtains pixel-wise information of an object-of-interest across multiple wavelengths of the electromagnetic spectrum from specialized hardware Manolakis et al. (2016). This produces an 'image cube' a 2D image with $C$ channels, as in qMRI, image channels correspond to the measurements. Experimental design involves choosing a design consisting of wavelengths and/or filters Arad & Ben-Shahar (2017); Waterhouse & Stoyanov (2022); Wu et al. (2019) which controls the image channels, and where most current practice uses uniform spacing for the wavelengths Thompson et al. (2017). For our paradigm, expensive devices can acquire large numbers of images with different spectral sensitivity simultaneously to provide training data for the design of much cheaper deployable devices Stuart et al. (2020). Recovering high-quality information from the few wavelengths chosen for particular applications by experimental design, reduces acquisition cost, increases acquisition speed, avoids misalignment, reduces storage requirements, and speeds up clinical adoption. Hyperspectral imaging has many applications Khan et al. (2018) from multiple modalities in medical imaging z Lu & Fei (2014); Karim et al. (2022), remote sensing Baumgardner et al. (2015), and environmental monitoring Stuart et al. (2019).

# F    DATA AND TASK EVALUATION

Table 13: Summary of data used in this paper.

| Name | Results | Data Type | Channels $\bar{C}$ | Target Regressors | Pixel/Voxel Size | No. Independent Variables | No. pixels/voxels $n$ x10$^3$ Train | Val | Test |
|---|---|---|---|---|---|---|---|---|---|
| VERDICT | Table 1 | Simulated qMRI | 220 | 8 | - | $\mathbf{d}^i \in \mathbb{R}^7$ | 1000 | 100 | 100 |
| NODDI | Table 7 | Simulated qMRI | 3612 | 7 | - | $\mathbf{d}^i \in \mathbb{R}^7$ | 100 | 10 | 10 |
| MUDI | Table 2 | qMRI Scan | 1344 | 1344 | 2.5mm$^3$ | $\mathbf{d}^i \in \mathbb{R}^6$ | 321 | 132 | 105 |
| HCP | Figure 2 | qMRI Scan | 288 | 288 | 1.25mm$^3$ | $\mathbf{d}^i \in \mathbb{R}^4$ | 2182 | 774 | 674 |
| Indian Pine | Table 3 | Remote Sensing Hyperspectral | 220 | 220 | 20m$^2$ | $\mathbf{d}^i \in \mathbb{R}$ | 1480 | 164 | 1135 |
| Oxygen Saturation | Table 4 | Simulated Hyperspectral | 348 | 2 | - | $\mathbf{d}^i \in \mathbb{R}^2$ | 0.4 | 0.044 | 10 |

This section includes additional details about the experimental data. Table 13 provides a summary, figure 6 visualizes various examples,figure 5 is a correlation plot of the measurements/channels/features.

We follow Grussu et al. (2021) and normalize each channel/measurement/feature by dividing by its $99th$ percentile value calculated from the training set. This is performed in both the input and output of the neural network.

Table 14: Parameter ranges for simulating synthetic VERDICT and NODDI model data.

| VERDICT Model | | | | NODDI Model | | |
|---|---|---|---|---|---|---|
| Parameter | Minimum | Maximum | | Parameter | Minimum | Maximum |
| $f_I$ | 0.01 | 0.99 | | $f_{ic}$ | 0.01 | 0.99 |
| $f_V$ | 0.01 | 0.99 | | $f_{iso}$ | 0.01 | 0.99 |
| $D_v$ ($\mu$ms$^2$ s$^{-1}$) | 3.05 | 10 | | $ODI$ | 0.01 | 0.99 |
| $R$ ($\mu$m) | 0.01 | 20 | | $\mathbf{n}$ | [-1 -1 -1] | [1 1 1] |
| $\mathbf{n}$ | [-1 -1 -1] | [1 1 1] | | | | |

## F.1 SIMULATIONS WITH THE VERDICT AND NODDI BIOPHYSICAL MODELS.

This section describes the the VERDICT and NODDI models and the experimental settings used in tables 1, 7. Exact code to perform the simulations is in Code Link.

The VERDICT (Vascular, Extracellular and Restricted Diffusion for Cytometry in Tumors) model Panagiotaki et al. (2014), maps histological features of solid-cancer tumors particularly for early detection and classification of prostate cancer Panagiotaki et al. (2015a); Johnston et al. (2019); Singh et al. (2022). The VERDICT model includes parameters: $f_I$ the intra-cellular volume fraction, $f_V$ the vascular volume fraction, $D_v$ the vascular perpendicular diffusivity, $R$ the mean cell radius, and $\mathbf{n}$ - a 3D vector defining mean local vascular orientation.

The NODDI (Neurite Orientation Dispersion and Density Imaging) model Zhang et al. (2012), maps parameters of the cellular composition of brain tissue and is widely used in neuroimaging studies in neuroscience such as the UK Biobank study Alfaro-Almagro et al. (2018), and neurology e.g. in Alzheimer's disease Kamiya et al. (2020) and multiple sclerosis Grussu et al. (2017). The NODDI model includes five tissue parameters: $f_{ic}$ the intra-cellular volume fraction, $f_{iso}$ the isotropic volume fraction, the orientation dispersion index (ODI) that reflects the level of variation in neurite orientation, and $\mathbf{n}$ - a 3D vector defining the mean local fiber orientation.

To conduct the simulations on the VERDICT and NODDI models, we employ the widely-used, open-source dmipy toolbox Fick et al. (2019). The code is available: Code Link. In each case, data simulation uses a known, fixed, acquisition scheme, i.e. experimental design, in combination with a set of ground truth model parameters. We chose the ground truth model parameters $\{\boldsymbol{\theta}_1, ..., \boldsymbol{\theta}_n\}$ for voxel/sample $i = 1, ..., n$ by uniformly sampling parameter combinations from the bounds given in table 14. We choose these bounds as they approximate the physically feasible limits of the parameters.

The VERDICT data has number of samples $n = 1000K, 100K, 100K$ in the train, validation, test split, with target data $Y \in \mathbb{R}^{n \times 8}, \boldsymbol{\theta}_i \in \mathbb{R}^8, i = 1, ..., n$. The classical experimental design approach yields an acquisition scheme derived from the Fisher information matrix Panagiotaki et al. (2015b) and here $X_D \in \mathbb{R}^{n \times 20}, C = 20$. The approaches in supervised feature selection (including TADRED) also use a densely-sampled empirical acquisition scheme, designed specifically for the VERDICT protocol from Panagiotaki et al. (2015a) and here $X_{\bar{D}} \in \mathbb{R}^{n \times 220}$ with $\bar{C} = 220$ measurements.

The NODDI data has number of samples $n = 100K, 10K, 10K$ in the train,validation,test split, with target data $Y \in \mathbb{R}^{n \times 7}, \boldsymbol{\theta}_i \in \mathbb{R}^7, i = 1, ..., n$. The classical experimental design approach yields an acquisition scheme derived from the Fisher information matrix Zhang et al. (2012) and so $X_{\bar{D}} \in \mathbb{R}^{n \times 99}, C = 99$. The approaches in supervised feature selection use a densely-sampled empirical acquisition scheme from an extremely rich acquisition from Ferizi et al. (2017). This was designed for the ISBI 2015 White Matter Challenge, which aimed to collect the richest possible data to rank biophysical models, and required a single subject to remain motionless for two uncomfortable back-to-back 4 hour scans. Here $X_{\bar{D}} \in \mathbb{R}^{n \times 3612}$ with $\bar{C} = 3612$ measurements.

We added Rician noise to all simulated signals, which is standard for MRI data Gudbjartsson & Patz (1995). The signal to noise ratio of the unweighted signal is 50, which is representative of clinical qMRI.

## F.2 MUlti-DIffusion (MUDI) CHALLENGE DATA

Data used in tables 2, 9 are images from five in-vivo human subjects, and are publicly available MUDI Organizers (2022), and was acquired with the state-of-the-art ZEBRA sequence Hutter et al. (2018). This diffusion-relaxation MRI dataset has a 6D acquisition parameter space $\mathbf{d}^i \in \mathbb{R}^6$:

echo time (TE), inversion time (TI), b-value, and b-vector directions in 3 dimensions: $b_x, b_y, b_z$. Data has 2.5mm isotropic resolution and field-of-view $220 \times 230 \times 140$mm and resulted in 5 3D brain volumes (i.e. images) with $\bar{C} = 1344$ measurements/channels, which here are unique diffusion- $T2^*$ and $T1$- weighting contrasts. More information is in Hutter et al. (2018); Pizzolato et al. (2020). Each subject has an associated brain mask, after removing outlier voxels resulted in $104520, 110420, 105743, 132470, 105045$ voxels for respective subjects $11, 12, 13, 14, 15$. For the experiment in table 2 we follow Blumberg et al. (2022) and perform 5-fold cross validation on the 5 subjects. For the experiment in table 9, we followed the original challenge Pizzolato et al. (2020) and took subjects $11, 12, 13$ as the training and validation set, and subjects $14, 15$ as the unseen test set, where $90\% - 10\%$ of the training/validation set voxels are respectively, for training and validation.

### F.3 HUMAN CONNECTOME PROJECT (HCP) TEST-RETEST DATA

This section describes the data and model fitting procedure used in figure 2 and table 8.

This section utilizes WU-Minn Human Connectome Project (HCP) diffusion data, which is publicly available at www.humanconnectome.org (Test Retest Data Release, release date: Mar 01, 2017) Essen et al. (2013). The data comprises $\bar{C} = 288$ volumes (i.e. measurements/channels), with 18 b=0 s mm$^{-2}$ (i.e. non-diffusion weighted) volumes, 90 gradient directions for b=1000 s mm$^{-2}$, 90 directions for b=2000 s mm$^{-2}$, and 90 directions for b=3000 s mm$^{-2}$. We used 3 scans for training (ID numbers $103818\_1, 105923\_1, 111312\_1$), one scan for validation ($114823\_1$) and one scan for testing ($115320\_1$), which produced numbers of samples $n = 708724 + 791369 + 681650 = 2181743, 774149, 674404$ for the respective splits. We used only voxels inside the provided brain mask and normalized the data voxelwise with a standard technique in MRI, by dividing all measurements by the mean signal in each voxel's b=0 values. Undefined voxels were then removed.

Diffusion tensor imaging (DTI) Basser et al. (1994), diffusion kurtosis imaging (DKI) Jensen & Helpern (2010), and Mean Signal DKI (MSDKI) Henriques (2018) are widely-used qMRI methods. Like NODDI and VERDICT, they use diffusion MRI to sensitize the image intensity to the Brownian motion of water molecules within the tissue to provide a window on tissue microstructure. However, whereas NODDI and VERDICT are designed specifically for application to brain tissue and cancer tumors, respectively, DTI and DKI are more general purpose techniques that provide indices of diffusivity (e.g. mean diffusivity - MD), diffusion anisotropy (e.g. fractional anisotropy Basser & Pierpaoli (1996) - FA), and the deviation from Gaussianity, or kurtosis, (e.g. mean kurtosis Jensen & Helpern (2010) - MK) that can inform on tissue integrity or pathology. Mean signal diffusion kurtosis imaging (MSDKI) is a simplified version of DKI that quantifies kurtosis using a simpler model that is easier to fit Henriques (2018). These techniques show promise for extracting imaging biomarkers for a wide variety of medical applications, such mild brain trauma, epilepsy, stroke, and Alzheimer's disease Jensen & Helpern (2010); Ranzenberger & Snyder (2022); Tae et al. (2018).

To fit the DTI, DKI, and MSDKI biophysical models to the data, and obtain the downstream metrics (parameter maps), we employ the widely-used, open-source DIPY library Garyfallidis et al. (2014). We followed standard practice for model fitting in MRI and used the least-squares optimization approach and default fitting settings. To remove outliers, values were clamped where DTI FA $\in [0, 1]$, DTI MD,AD,RD $\in [0, 0.003]$, DKI MK,AK,RK $\in [0, 3]$, MSDKI MSD $\in [0, 0.003]$ MSDKI MSK $\in [0, 3]$. Code for model fitting is in Code Link.

The results in figure 2 and table 8 are all scaled by: DTI-FA $\times 10^2$, DTI-MD $\times 10^9$, DTI-AD $\times 10^9$, DTI-RD $\times 10^9$, DKI-MK $\times 10^2$, DKI-AK $\times 10^2$, DKI-RK $10^2$, MSDKI-MSD $\times 10^9$, MSDKI-MSK $\times 10^2$.

### F.4 AIRBORNE VISIBLE / INFRARED IMAGING SPECTROMETER (AVIRIS) DATA AND TASK

This section describes the data and task considered in tables 3, 10.

The Airborne Visible / Infrared Imaging Spectrometer (AVIRIS) is a highly-specialized hyperspectral device for earth remote sensing commissioned by the Jet Propulsion Laboratory (JPL). It obtains acquisitions from adjacent spectral channels bands between the wavelengths 400nm - 2500nm. It is flown from four different aircrafts and has been deployed worldwide, for purposes such as examining the effect and rehabilitation of forests affected by large wildfires, the effect of climate change, and

other applications in atmospheric studies and snow hydrology. More information is available Jet Propulsion Laboratory (JPL) (2023); Simmonds & Green (1996); Thompson et al. (2017) and on the webpage `https://aviris.jpl.nasa.gov`.

Data used was obtained in June 1992, when the Purdue University Agronomy Department commissioned AVIRIS to obtain two ground images of the 'Indian Pine' to support soils research Baumgardner et al. (2015) from two flight lines: east-to-west and north-to-south. This is publicly available Baumgardner et al. (2022). The data are two 'image cube' corresponding to a 2miles$^2$ area of 20m$^2$ pixel size with $\bar{C} = 220$ channels.

Data from the north-to-south flight are used for training and validation. This consists of 1644292 pixels of which 90-10 % were used for training-validation. Data from the east-to-west flight were used for test data, which consists of 1134672 pixels. We removed outliers from both images - details in Code Link and then normalized the image channel-wise so the $99th$-percentile is $255$ (the maximum in standard images).

The objective is examine whether our supervised feature selection approaches can reconstruct the entire image from a subset of wavelengths, typical of the ground obtained over Indiana (the location of 'Indian Pine').

## F.5 ESTIMATION OF OXYGEN SATURATION DATA AND TASK

This experiment and data follows directly from Waterhouse & Stoyanov (2022). Data was generated from the code presented in Waterhouse & Stoyanov (2022), with assistance from its author.

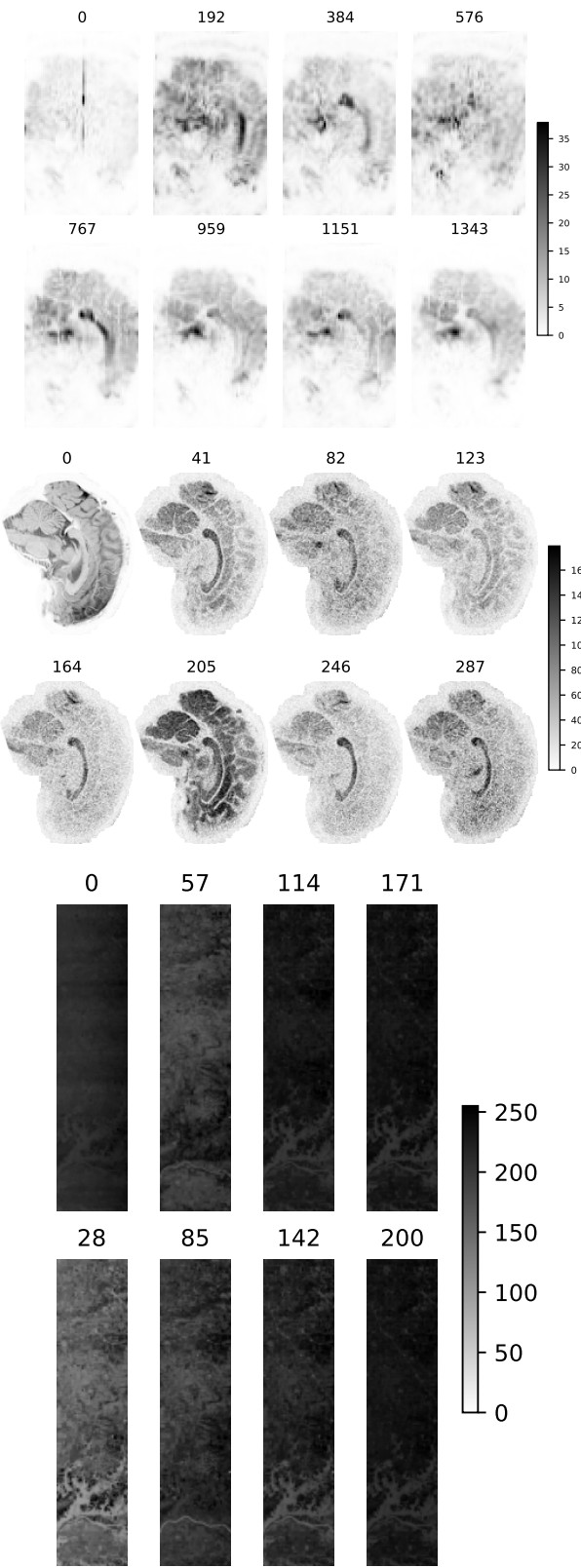

Figure 6: 2D brain slices from 3D MRI scans, for different measurements/features/channels (values of $C$) for the Multi-Diffusion (MUDI) challenge (top), HCP (middle) data. Bottom: Different wavelengths for the 'Indian Pine' remote sensing hyperspectral data north-to-south flight.

