# OpenReview forum: "Experimental Design for Multi-Channel Imaging via Task-Driven Feature Selection"
_ICLR.cc/2024/Conference — ICLR 2024 poster_

### Official Review · Reviewer_e6jZ · 2023-10-13

**Soundness:** 3 good
**Presentation:** 3 good
**Contribution:** 3 good
**Rating:** 6
**Confidence:** 2

**Summary:**

This work suggests a task-driven paradigm for experimental design for real-world imaging applications, that does not require a-priori model specification and replaces high-dimensional continuous search with a subsampling problem.

**Strengths:**

+ Well written paper.
+ Good structure for Intro, related work and methods.
+ The problem setting and contributions are clearly communicated.
+ Comprehensive previous work and comparison to baselines.

**Weaknesses:**

- Hard to follow the results and it feels like they are rashly presented. I would reiterate table 1,2 and fig 2 and explain better what are the columns rows etc. Consider taking some information from appendix to the main text.
- Also you repeat some of the results from the beginning of section 4 afterwards in different paragraphs, which makes it hard to follow. I would reiterate here.

**Questions:**

- To me it is still not so clear the difference between feature selection and experimental design optimisation.In paper you mention the difference about features being irrelevant and channels being repetitive. How do you ensure or balance this in your work? Is there a specific loss you use?

---

> ### Author Response · Authors · 2023-11-19
> **Response to Reviewer e6jZ 1**
>
> Thank you for your review, we are pleased you thought 'Well written paper', 'Good structure for Intro, related work and methods.', 'The problem setting and contributions are clearly communicated.', and 'Comprehensive previous work and comparison to baselines.'
>
> We address your points below and modify the paper - please consider raising your score if you have no further requests.
>
> > Hard to follow the results and it feels like they are rashly presented. I would reiterate table 1,2 and fig 2 and explain better what are the columns rows etc. Consider taking some information from appendix to the main text.
>
> Thank you for the comment.  We have added additional explanation to the tables and figures that you mention.
>
> > Also you repeat some of the results from the beginning of section 4 afterwards in different paragraphs, which makes it hard to follow. I would reiterate here.
>
> Thank you for this feedback.  The aim of the paragraph at the start of section 4 is to give a short overview of the experiments and results and how they fit together. We believe this is important signposting to help the reader navigate the extensive results section and understand how they fit together to demonstrate TADRED's overall benefit.  Then we go into detail for each experiment in each paragraph.  We're unsure what you are suggesting we reiterate  and where, but we are keen to make this large section as clear as possible - any specific advice gratefully received!
>
> > To me it is still not so clear the difference between feature selection and experimental design optimisation. In paper you mention the difference about features being irrelevant and channels being repetitive. How do you ensure or balance this in your work? Is there a specific loss you use?}
>
> In general the problems of experiment design and feature selection differ in that the former is usually continuous (identifying a combination of points in a space of possible measurement settings) -- see first paragraph and equation 1 whereas the latter is inherently discrete (selecting a subset of a predefined superset) -- see paragraphs around equation 2.  The novelty of the paradigm we propose is to equate the two tasks  As we explain in paragraph 6 of the introduction, the key difference in the problems once equated in this way lies in the structure of the problem: often in classical feature selection problems most features are uninformative and the challenge is to identify a small fraction that carry important information.  In the experiment design problem, most features/channels are informative, as it is usually easy to identify image channels with no information content a priori.  The challenge is to identify a combination of channels that are maximally complementary.  This difference demands a different search strategy, which we expand on in `Supervised Feature Selection' paragraph of the related work.  TADRED does not require a particular loss, similar to the baselines, we use mean-squared-error (MSE).

---

> ### Author Response · Authors · 2023-11-22
> **End of Author-Reviewer Discussion**
>
> We greatly appreciate the time and effort you’ve dedicated to reviewing the paper. We’ve carefully considered your feedback and have updated the manuscript accordingly - we kindly request that you reconsider your initial score in light of these revisions. Please let us know if there are any further clarifications that we can offer before the end of the author-reviewer discussion period.

---

### Official Review · Reviewer_aeL1 · 2023-10-31

**Soundness:** 3 good
**Presentation:** 4 excellent
**Contribution:** 3 good
**Rating:** 6
**Confidence:** 3

**Summary:**

This paper proposes a new approach to Experimental Design in imaging tasks, where the goal is to select a restricted set of channels from multi-channel images to solve a given task (e.g. reconstruction of missing channels).
The approach relies on 2 networks optimized jointly: a score-network and a task network. Throughout optimization, channels are dropped at regular intervals based on the score-network's average output for these channels.
Experiments on several datasets are then conducted to show the efficiency of the approach.

**Strengths:**

- **Code**: The code is present and allows reproducing the results.
- **Experiments**: The experimental section shows impressive results and must have required a huge amount of work to benchmark all these datasets and all these methods.
- **Prior work**: The prior work is abundantly discussed in relation to the proposed approach which makes for a great read and gives a lot of interesting context to understand the relevance of the different implementation choices.
- **Clarity**: More generally, I would say the paper is well written and clear, with particularly well-worked figures.

**Weaknesses:**

- **Validation of score network**: I think a nice validation that the score network is actually working as intended would be to set random score values at the beginning of optimization of TADRED for each channel (and keep them fixed). This way, the task network is optimized for the same number of steps and in the same manner, i.e. with gradually less channels. Maybe this gradual aspect is actually making the reconstruction network better and it's why TADRED is performing better. I list this as a weakness and not as a question, because to me the experiments are lacking of proof that the score network is working as intended and not just a smart preprocessor or anything else.
- **Experiments**: my current problem with the experimental section is that I see it in 2 folds: qMRI/EOS where TADRED outperforms state-of-the-art | others where TADRED outperforms baselines. In the others setting, I gather that there was no previous interest reported in the literature for solving the task with ED (i.e. no published results to compare against). For qMRI/EOS, I have an issue because it seems that these 2 tasks have not gathered a lot of interest either if we simply look at the citation count for the papers introducing the challenges (17 for MUDI and 5 for EOS).

This last reason is why I am not increasing my score at the moment.

**Questions:**

- I see that random feature selection sometimes performs on par with other methods especially in high number of selected channels regimes. Why is that?

---

> ### Author Response · Authors · 2023-11-23
> **Response to Reviewer aeL1 Part 1**
>
> Thank you for the review.  Profuse apologies for the delay.  We originally misinterpreted your request and required additional time - we thank you for your patience.
>
> We are pleased you liked the code, the extensive experiments, the relationship of TADRED with prior work, and thought our paper was clear.  Indeed, a lot of work has gone into constructing these demonstrations, we are very pleased you appreciate the efforts.
>
> > Validation of score network: I think a nice validation that the score network is actually working as intended would be to set random score values at the beginning of optimization of TADRED for each channel (and keep them fixed). This way, the task network is optimized for the same number of steps and in the same manner, i.e. with gradually less channels. Maybe this gradual aspect is actually making the reconstruction network better and it's why TADRED is performing better. I list this as a weakness and not as a question, because to me the experiments are lacking of proof that the score network is working as intended and not just a smart preprocessor or anything else.
>
> Thank you for the question.  We have now tested a modified training procedure that works in the 'same manner' whilst the scores are random. In settings of table 5 results (MSE) are more than 10 \% worse, even more than 'w/o iterative subsampling' - we are repeating over more randomizations before adding any statement to the paper.  Thus, as you suggest, although the 'gradual aspect' you mention improves performance (in fact line 2 in the ablation study in table 5 already demonstrates this with `less gradual' scenario without iterative subsampling decreases performance), the learning of the scoring network is working as intended and further improves results. The objective is (see related work) for the 'scoring and subsampling procedure enables efficient identification of subsets of complementarily informative channels jointly with training a high-performing network for the task' -- which is showed through large improvements over the baselines in multiple tasks.

---

### Official Review · Reviewer_FaxL · 2023-11-01

**Soundness:** 3 good
**Presentation:** 2 fair
**Contribution:** 3 good
**Rating:** 6
**Confidence:** 3

**Summary:**

The paper presents a new task-specific scheme for experimental design for imaging data and application. The authors focus on expanding the design to go beyond and aim at user-specific analysis tasks at the same time. The authors proposed the TAsk-DRiven experimental design in imaging, or TADRED, to select the channel-subset while at the same time carry out the tasks.

**Strengths:**

1. The case studies are quite extensive and substantial, showing the performance of the proposed framework against a variety of applications in feature selection.

**Weaknesses:**

1. In the experiment section, although quite a number of baselines are used for comparison, including Fisher-matrix, the baselines are not clearly explained. Some details are included in the appendix, but it may be better to at least explain one of them in detail.

**Questions:**

N/A

---

> ### Author Response · Authors · 2023-11-19
> **Response to Reviewer FaxL 1**
>
> Thank you for your review, we are pleased you thought the 'case studies are quite extensive and substantial, showing the performance of the proposed framework against a variety of applications in feature selection'.  Please consider updating your score if you have no further comments.
>
> > In the experiment section, although quite a number of baselines are used for comparison, including Fisher-matrix, the baselines are not clearly explained. Some details are included in the appendix, but it may be better to at least explain one of them in detail.
>
> Thank you for the feedback.
>
> The baselines are quite complicated.  For example FIRDL's pseudocode in [93]-appendix-D runs to 1.5 pages and SSEFS [56]-algs-1,2,3,4 is even longer.  In fact we believe the relative simplicity of our algorithm is a nice advantage of our method.  Thus, rather than explain in great detail, we have tried to give the essence of each method in related work section 2 and refer to the corresponding papers for full descriptions.  Note Wojtas \& Chen (2020) and Lee et al. (2022) -- and others use the same approach when discussing prior work / baselines in the respective papers.  We do give substantial detail in Appendix A and while we don't think it is useful to repeat the full detail of any particular method if the reviewer feels there is necessary information missing we would be very happy to add it.
>
> Please note we discussed the baselines and how they relate to our approach in the related work section 2 which is quite expansive (greater than 1 page). More specifically the Fisher Information Baseline (used in table 1 top, table 7 top) in 'Approaches in Experimental Design'; and the supervised feature selection baselines (FIRDL, SSEFS) along with the PROSUB Blumberg et al. (2022) in the three paragraphs of `Supervised Feature Selection'.  Finally, the other reviewers liked the discussion of prior work (see general response).

---

> ### Author Response · Authors · 2023-11-22
> **End of Author-Reviewer Discussion**
>
> We greatly appreciate the time and effort you’ve dedicated to reviewing the paper.
> We’ve carefully considered your feedback we kindly request that you reconsider your initial score in light of these revisions.
> Please let us know if there are any further clarifications that we can offer before the end of the author-reviewer discussion period.

---

### Official Review · Reviewer_Kzed · 2023-11-01

**Soundness:** 3 good
**Presentation:** 2 fair
**Contribution:** 2 fair
**Rating:** 6
**Confidence:** 2

**Summary:**

This paper develops an automated approach for channel subsampling and successfully applies this apporach to several MRI and hyperspectral imaging applications. The approach, TADRED, combines existing feature selection literature with recursive feature elimination (RFE). In particular, TADRED gradually reduces the number of features/channels used. Each time it reduces the feature/channel set it solves an optimization problem (3) that simultaneously trains a network to solve a task, designs a binary mask (optimized as continuous numbers) that subsamples the remaining channels, and learns a weighting matrix that weights the remaining channels fed into the network. TADRED outperforms several baselines on 6 distinct tasks.

**Strengths:**

Proposed method is validated across many distinct tasks

Proposed methods seems to outperform existing baselines by a small margin.

Selecting a subset of informative channels is an interesting problem.

**Weaknesses:**

The paper could do a better job explaining the proposed method. For instance, algorithm 2 would benefit from additional comments. I have little intuition for what each step is trying to do.

The figures table captions are generally uninformative which makes the results very difficult to follow:
-In figure 1, "Model Free" is uninformative.
-Table 1 is very hard to parse. It's essentially two tables, with the top table having a different format than the bottom one.
-Table 2 doesn't mention MRI anywhere.
-Figure 2 is covered with arrows, but the caption doesn't state what they're pointing to.

"Feature Fill" is not defined in the main text

**Questions:**

##  Minor comments
The sentence, "e.g. the standard design for VERDICT model Panagiotaki et al. (2015a) (used as a baseline in table 1) is computed by optimizing the Fisher-matrix for one specific combination of parameter values, despite aiming to highlight contrast in those parameters throughout the entire prostate" in related work is provided without the context of prostate cancer.

The statement "Rather than learning the mask mt end-to-end e.g. using a sparsity term/prior, we modify elements of mt during our training procedure" could use further clarification

The baseline methods were all published in medical imaging journals. This work might be better appreciated, and receive more informed reviews, in such a venue.

---

> ### Author Response · Authors · 2023-11-19
> **Respond to Reviewer Kzed 1**
>
> Thank you for taking the time to review the paper and engaging with the submission. We are pleased you appreciate our multiple applications and found our problem interesting.  We address all your points below  and upload a new pdf with the changes.  Please consider raising your score if you are satisfied and do not have further questions.
>
> > Proposed methods seems to outperform existing baselines by a small margin
>
> Please note this comment was under `strengths'.  Actually the margins are often substantial, for example, average of 15.7 \% improvement across 8 subsets for MRI in table 2, average of 25 \% in hyperspectral table 3.
>
> > The paper could do a better job explaining the proposed method. For instance, algorithm 2 would benefit from additional comments. I have little intuition for what each step is trying to do.
>
> Thank you for highlighting this.  The intention was to keep the intuition to the main text leaving the algorithm succinct and uncluttered.  We have now added comments to the algorithm to make the mapping between steps of the algorithm and blocks of the main text clear, which we hope resolves this issue.
>
> **The figures table captions are generally uninformative which makes the results very difficult to follow**
>
> > In figure 1, "Model Free" is uninformative.
>
> We agree and have removed `model free' from the figure.
>
> > Table 1 is very hard to parse. It's essentially two tables, with the top table having a different format than the bottom one.
>
> Thank you for the suggestion.  As the top table only has a single comparison, we replaced it with a sentence in the main text (paragraph `TADRED Outperforms Classical Experimental Design and Baselines in Model Parameter Estimation').
>
> > Table 2 doesn't mention MRI anywhere
>
> Thank you for the recommendation, we expanded the caption (and added `MRI') to the updated document.
>
> > Figure 2 is covered with arrows, but the caption doesn't state what they're pointing to
>
> Thank you for spotting this.  We have adjusted the caption accordingly `Right: Qualitative comparison where arrows highlight closer agreement from TADRED's design with the gold standard than those from the best performing baseline in each metric.'
>
> > "Feature Fill" is not defined in the main text
>
> Thank you for pointing this out.  We have explicitly added `feature fill' in the text under equation (3) page 5 where the corresponding symbol first appears.
>
> **Minor comments**
>
> > The sentence, "e.g. the standard design for VERDICT model Panagiotaki et al. (2015a) ... highlight contrast in those parameters throughout the entire prostate" in related work is provided without the context of prostate cancer.
>
> We modified the main text and adaded `with primary application in prostate cancer detection and classification'.
>
> > The statement "Rather than learning the mask mt end-to-end e.g. using a sparsity term/prior, we modify elements of mt during our training procedure" could use further clarification
>
> We are referring to the baseline SSEFS approach here, we have modified the text to include an explicit reference.  Please note in the `Supervised Feature Selection' paragraph in section 2 we describe the relationship of TADRED with the SSEFS baseline in detail.
>
> > The baseline methods were all published in medical imaging journals. This work might be better appreciated, and receive more informed reviews, in such a venue
>
> We believe ICLR is the right venue for this work. Most experiment design methods papers, which this is, are in statistics or machine learning venues e.g.  Blau et al. (2022); Foster et al. (2021); Ivanova et al. (2021). As you say it does present an "interesting problem" for those communities. It is true that applications of those methods are generally in applications journals: the MRI applications papers we refer to are indeed mostly in medical imaging journals, the hyperspectral imaging ones, such as our results in table 4, are actually mostly in the optics literature.  However, for example, the feature selection baselines FIRDL [Wojtas & Chen (2020)], SSEFS [Lee
> et al. (2022)] used in tables 1, 6, 7 and figure 2 were published in machine learning conferences: NeurIPS and ICLR.
>
> TADRED is a method for experiment design rather than an application in a specific area, so we believe ICLR is appropriate.  The technique extends to many further multi-channel applications and potentially beyond multi-channel imaging -- see appendix F and the discussion section 5.

---

> ### Author Response · Authors · 2023-11-22
> **End of Author-Reviewer Discussion**
>
> We greatly appreciate the time and effort you’ve dedicated to reviewing the paper.
> We’ve carefully considered your feedback and have updated the manuscript accordingly - we kindly request that you reconsider your initial score in light of these revisions.
> Please let us know if there are any further clarifications that we can offer before the end of the author-reviewer discussion period.

---

### Author Response · Authors · 2023-11-19
**General Response**

We are grateful to all the reviewers for taking the time to read our paper and providing feedback.

We are happy to see that all reviewers recommend acceptance.

We are pleased that the reviewers appreciate our multiple applications: 'proposed method is validated across many distinct tasks' (R-Kzed), 'case studies are quite extensive and substantial' (R-FaxL); clarity and organization: 'paper is well written and clear, with particularly well-worked figures' (R-aeL1), 'well written paper' with 'good structure for intro, related work and methods' (R-e6jZ); relation with prior work: 'prior work is abundantly discussed in relation to the proposed approach which makes for a great read and gives a lot of interesting context to understand the relevance of the different implementation choices' (R-aeL1), Comprehensive previous work and comparison to baselines (R-e6jZ) and the code: 'code is present and allows reproducing the results' (R-aeL1).

We address all of the reviewers' concerns below and have updated the pdf of our submission. Changes in the main submission that reflect reviewers' comments are in red text, we also made small changes to formatting.

We look forward to interacting with the reviewers.

---

### Meta-Review · Area_Chair_gisT · 2023-12-14

**Metareview:**

This paper presents a feature subselection method for optimizing imaging systems for specific tasks. Features are selected along with, and relative to, a back-end deep network. This "TAsk DRiven Experimental Design" (TADRED) is demonstrated on MRI and Hyperspectral imaging data. The reviews were primarily positive, with the majority of comments stemming from, or related to the clarity of the model, problem, and results. However, the reviewers did note the well-done code/data for reproducibility and otherwise noted no major weaknesses. Therefore I recommend this paper be accepted.

**Justification For Why Not Higher Score:**

The overall reception was lukewarm, which may be related to the many clarity-based comments from the reviewers.

**Justification For Why Not Lower Score:**

The overall positive scores, with the positive notes on reproducibility, are pluses for this work.

---

### Decision · Program_Chairs · 2024-01-16

Accept (poster)